# Machine intelligence accelerated design of conductive MXene aerogels with programmable properties

Snehi Shrestha[1], Kieran James Barvenik[2], Tianle Chen [1], Haochen Yang [1], Yang Li[1], Meera Muthachi Kesavan[1], Joshua M. Little[1], Hayden C. Whitley[1], Zi Teng[3], Yaguang Luo[3], Eleonora Tubaldi [2,4] ✉ & Po-Yen Chen [1,4] ✉

Designing ultralight conductive aerogels with tailored electrical and mechanical properties is critical for various applications. Conventional approaches rely on iterative, time-consuming experiments across a vast parameter space. Herein, an integrated workflow is developed to combine collaborative robotics with machine learning to accelerate the design of conductive aerogels with programmable properties. An automated pipetting robot is operated to prepare 264 mixtures of $Ti_3C_2T_x$ MXene, cellulose, gelatin, and glutaraldehyde at different ratios/loadings. After freeze-drying, the aerogels' structural integrity is evaluated to train a support vector machine classifier. Through 8 active learning cycles with data augmentation, 162 unique conductive aerogels are fabricated/characterized via robotics-automated platforms, enabling the construction of an artificial neural network prediction model. The prediction model conducts two-way design tasks: (1) predicting the aerogels' physico-chemical properties from fabrication parameters and (2) automating the inverse design of aerogels for specific property requirements. The combined use of model interpretation and finite element simulations validates a pronounced correlation between aerogel density and compressive strength. The model-suggested aerogels with high conductivity, customized strength, and pressure insensitivity allow for compression-stable Joule heating for wearable thermal management.

Conductive aerogels have gained significant research interests due to their ultralight characteristics, adjustable mechanical properties, and outstanding electrical performance[1–6]. These attributes make them desirable for a range of applications, spanning from pressure sensors[7–10] to electromagnetic interference shielding[11–13], thermal insulation[14–16], and wearable heaters[17–19]. Conventional methods for the fabrication of conductive aerogels involve the preparation of aqueous mixtures of various building blocks, followed by a freeze-drying process[20–23]. Key building blocks include conductive nanomaterials like carbon nanotubes, graphene, $Ti_3C_2T_x$ MXene nanosheets[24–30], functional fillers like cellulose nanofibers (CNFs), silk nanofibrils, and chitosan[29,31–34], polymeric binders like gelatin[25,26], and crosslinking agents that include glutaraldehyde (GA) and metal ions[30,35–37]. By adjusting the proportions of these building blocks, one can fine-tune the end properties of the conductive aerogels, such as electrical conductivities and compression resilience[38–41]. However, the correlations

[1]Department of Chemical and Biomolecular Engineering, University of Maryland, College Park, MD 20742, USA. [2]Department of Mechanical Engineering, University of Maryland, College Park, MD 20742, USA. [3]US Department of Agriculture, Agricultural Research Service, Food Quality Laboratory and Environment Microbial Food Safety Laboratory, Beltsville Agricultural Research Center, Beltsville, MD 20725, USA. [4]Maryland Robotics Center, College Park, MD 20742, USA. ✉e-mail: etubaldi@umd.edu; checp@umd.edu

between compositions, structures, and properties within conductive aerogels are complex and remain largely unexplored[42–47]. Therefore, to produce a conductive aerogel with user-designated mechanical and electrical properties, labor-intensive and iterative optimization experiments are often required to identify the optimal set of fabrication parameters. Creating a predictive model that can automatically recommend the ideal parameter set for a conductive aerogel with programmable properties would greatly expedite the development process[48].

Machine learning (ML) is a subset of artificial intelligence (AI) that builds models for predictions or recommendations[49–51]. AI/ML methodologies serve as an effective toolbox to unravel intricate correlations within the parameter space with multiple degrees of freedom (DOFs)[50,52,53]. The AI/ML adoption in materials science research has surged, particularly in the fields with available simulation programs and high-throughput analytical tools that generate vast amounts of data in shared and open databases[54], including gene editing[55,56], battery electrolyte optimization[57,58], and catalyst discovery[59,60]. However, building a prediction model for conductive aerogels encounters significant challenges, primarily due to the lack of high-quality data points. One major root cause is the lack of standardized fabrication protocols for conductive aerogels, and different research laboratories adopt various building blocks[35,40,46]. Additionally, recent studies on conductive aerogels focus on optimizing a single property, such as electrical conductivity or compressive strength, and the complex correlations between these attributes are often neglected to understand[37,42,61–64]. Moreover, as the fabrication of conductive aerogels is labor-intensive and time-consuming, the acquisition rate of

training data points is highly limited, posing difficulties in constructing an accurate prediction model capable of predicting multiple characteristics.

Herein, we developed an integrated platform that combines the capabilities of collaborative robots with AI/ML predictions to accelerate the design of conductive aerogels with programmable mechanical and electrical properties (see Supplementary Fig. 1 for the robot–human teaming workflow). Based on specific property requirements, the robots/ML-integrated platform was able to automatically suggest a tailored parameter set for the fabrication of conductive aerogels, without the need for conducting iterative optimization experiments. To produce various conductive aerogels, four building blocks were selected, including MXene nanosheets, CNFs, gelatin, and GA crosslinker (see Supplementary Note 1 and Supplementary Fig. 2 for the selection rationale and model expansion strategy). Initially, an automated pipetting robot (i.e., OT-2 robot) was operated to prepare 264 mixtures with varying MXene/CNF/gelatin ratios and mixture loadings (i.e., solid contents in the mixtures), and these mixtures underwent a freeze-drying process to produce various aerogels. Based on the structural integrity and monolithic nature, these aerogels were categorized to train a support vector machine (SVM) classifier, and then a feasible parameter space was successfully defined. Next, through 8 active learning loops with data augmentation, 162 kinds of conductive aerogels were stagewise fabricated/characterized, and these data were input to construct an artificial neural network (ANN) model with high prediction accuracy. During the active learning loops, the data acquisition rate was increased by integrating an OT-2 robot and a UR5e

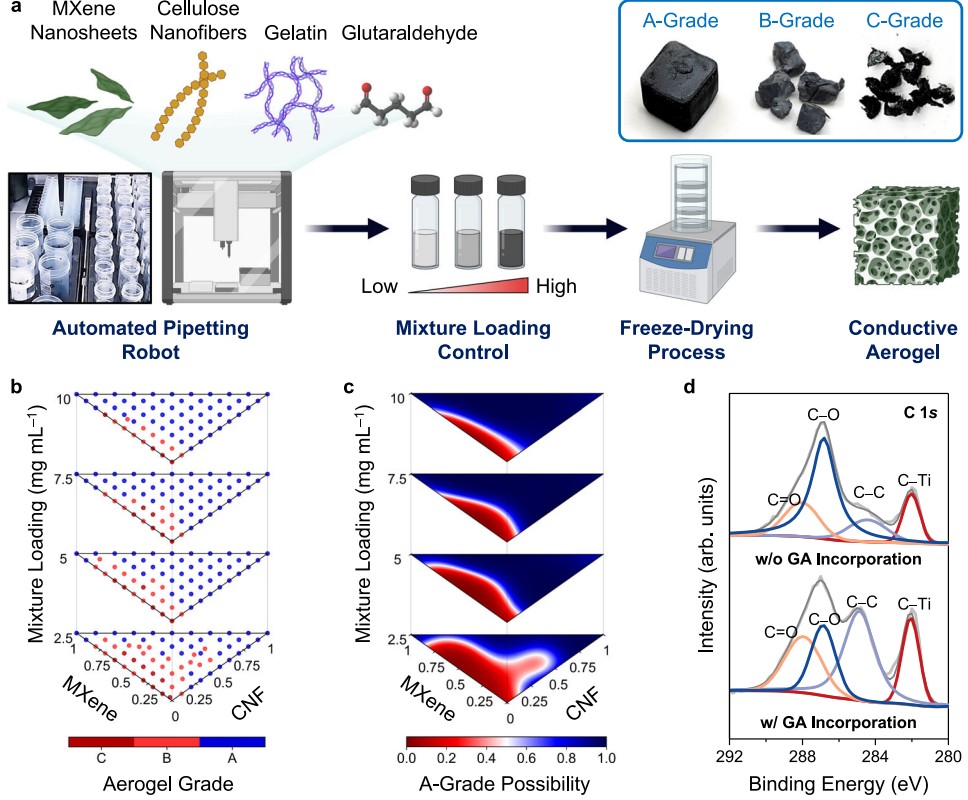

**Fig. 1 | Defining a feasible parameter space through automated pipetting robot and support-vector machine (SVM) classifier. a** Schematic illustration of the fabrication process of conductive aerogels accelerated by an automated pipetting robot (i.e., OT-2 robot). Four building blocks were incorporated, including MXene nanosheets, cellulose nanofibers (CNFs), gelatin, and glutaraldehyde (GA). By adjusting the MXene/CNF/gelatin/GA ratios and the mixture loadings (i.e., solid contents of aqueous mixtures), the mechanical and electrical properties of conductive aerogels were controlled. **b** 264 MXene/CNF/gelatin aerogels with different grades based on their structural integrity and monolithic nature. **c** Four heatmaps showcasing the possibilities of producing A-grade conductive aerogels at specific MXene/CNF/gelatin ratios and mixture loadings. **d** C 1$s$ XPS spectra of two MXene/CNF aerogels (at the 80/20 ratio and 10 mg mL$^{-1}$) with and without the GA incorporation.

collaborative robotic arm in the aerogel fabrication and characterization processes. By harnessing the model's prediction capabilities, two-way design tasks were realized: (1) accurately predicting the mechanical and electrical properties of a conductive aerogel based on a set of fabrication parameters, and (2) automatically discovering suitable conductive aerogels that satisfy specific property requirements. Through SHapley Additive exPlanations (SHAP) model interpretation, several data-driven insights were identified, and the pronounced impact of mixture loading on the aerogel's compressive strength was validated using Finite Element (FE) simulations. As a final demonstration, the prediction model was employed to discover a strain-insensitive conductive aerogel suitable for wearable thermal management. The model-suggested conductive aerogel exhibited high electrical conductivity, customized compression resilience, and ultralow pressure sensitivity, enabling efficient Joule heating performance upon repetitive compression cycles. Our hybrid approach, which seamlessly integrated robot-assisted experiments with AI/ML algorithms and simulation tools, not only enabled efficient customization of conductive aerogels but also provided a versatile workflow for other nanoscience fields[48,65–67].

## Results

### Tuning mechanical and electrical properties of conductive aerogels through varying multiple fabrication parameters

As shown in the TEM images in Supplementary Fig. 3a, b, $Ti_3C_2T_x$ MXene nanoshewets showed average lateral dimensions of $1.4 \times 1.1\,\mu m^2$, and CNFs exhibited an average diameter of 15 nm and an average length of $1\,\mu m$. Supplementary Fig. 4a reveals that CNF and MXene dispersions exhibited average zeta potentials of less than −40 mV. As shown in Supplementary Fig. 4a, b, the zeta potentials of MXene and CNF dispersions were monitored before and after 2-week of storage. Their zeta potentials remained consistent, with no signs of oxidation or aggregation detected. Supplementary Fig. 4c indicates that the MXene/CNF/gelatin/GA mixtures, across various ratios and mixture loadings, retained high dispersity. After undergoing a freeze-drying process at −80 °C and 0.3 Pa, these MXene/CNF/gelatin/GA mixtures at different ratios and mixture loadings produced various conductive aerogels. By modifying the MXene/CNF/gelatin/GA ratios and altering the mixture loadings, Supplementary Fig. 5a reveals a non-linear variation in the mechanical properties of conductive aerogels. Similarly, Supplementary Fig. 5b depicts the non-linear shifts in the electrical properties of conductive aerogels, such as electrical resistance, in response to changes in the fabrication parameters.

To establish a comprehensive database linking the fabrication parameters with the end properties of conductive aerogels, over 5300 data points are required, given a step size of 2.0 wt% and four mixture loadings (see our estimation in Supplementary Note 2 and Supplementary Fig. 6). However, building such a dataset is impractical due to time and resource constraints. To overcome this challenge, an integrated platform that leveraged collaborative robotics and AI/ML predictions was developed to acquire high-quality data points and construct a high-accuracy prediction model. By harnessing the model's prediction capabilities, the development of conductive aerogels with user-designated mechanical and electrical properties was facilitated.

### Defining a feasible parameter space through automated pipetting robot and support-vector machine (SVM) classifier

To construct a high-accuracy prediction model, an AI/ML framework was developed and had three critical phases: (1) establishing a feasible parameter space, (2) implementing active learning loops, and (3) synthesizing virtual data points. The rationale of each phase is detailed in Supplementary Note 3. Supplementary Table 1 summarized the descriptors (i.e., labels) used in the prediction model. Supplementary Fig. 7 compares the prediction accuracy of models using different labels to represent the mechanical properties of conductive aerogels.

As illustrated in Fig. 1a, the first phase aimed to define a feasible parameter space using an OT-2 robot and an SVM classifier. The OT-2 robot was commanded to prepare a library of aqueous mixtures with different MXene/CNF/gelatin ratios and mixture loadings (from 2.5 to $10.0\,mg\,mL^{-1}$). Supplementary Movie 1 showcases the OT-2 robot's efficiency to prepare 264 mixtures in 6 h, with an interval of 10 wt.% for four mixture loadings. Once prepared, these mixtures were vortexed, cast into silicone molds, and then subjected to a freeze-drying process. Afterward, the 264 freeze-dried samples were obtained and then categorized based on their structural integrity and monolithic nature (classification standards in Supplementary Note 4). As shown in the inset of Fig. 1a, classification varied from (1) sizable, intact samples (A-grade), (2) smaller fragmented samples (B-grade), to (3) extensively altered forms with inconsistent pieces (C-grade). As detailed in Fig. 1b and Supplementary Table 2, the collection consisted of 201 A-grade, 49 B-grade, and 14 C-grade samples. Two blind tests were performed by different researchers to maintain unbiased evaluations.

Afterward, these discrete grades served as training data points for a SVM classifier, the goal of which was to distinguish the hyperplanes with maximal margins between the data points at different grades (see Supplementary Note 5). Given a specific MXene/CNF/gelatin ratio, the trained SVM classifier was able to predict the possibility of obtaining an A-grade aerogel at a high accuracy of 95% (examined by a set of testing data points not previously introduced to the SVM classifier, Supplementary Table 3). As shown in Fig. 1c, the SVM classifier produced four heatmaps (one for each mixture loading), illustrating the possibilities of obtaining A-grade aerogels across the entire parameter space. By setting the A-grade possibility threshold at 65%, a feasible parameter space was defined in Supplementary Fig. 8a. Supplementary Fig. 8b shows that the area of the feasible parameter space shrank from 83.7% to 48.6%, as the mixture loading decreased from 10.0 to $2.5\,mg\,mL^{-1}$, respectively. Supplementary Fig. 9 illustrates similar data distribution plots using the MXene and gelatin loadings as the axes. Within the AI/ML framework, the SVM classifier acted as an important filtering unit for the prediction model, and only the MXene/CNF/gelatin ratios that led to A-grade aerogel production were suggested. The SVM classifier effectively eliminated the need of exploring of the unfeasible regions that led to fragile conductive aerogels with scale-up difficulties.

During the robot-assisted mixture preparation, an optional step is to incorporate GA as a crosslinking agent into the MXene/CNF/gelatin mixtures. GA is widely acknowledged as a chemical crosslinker for CNFs[68], gelatin[69], and MXene nanosheets[30,36]. To investigate possible covalent bonds formed between CNFs and/or MXene nanosheets, two kinds of MXene/CNF aerogels (at the 80/20 ratio and at $10\,mg\,mL^{-1}$) were produced: one incorporated with GA (denoted as "+") and the other without GA (as "−"). Afterward, X-ray photoelectron spectroscopy (XPS) was adopted to characterize the chemical bonds in two MXene/CNF aerogels. As shown in Fig. 1d, the C 1 s spectrum of the GA-crosslinked aerogel demonstrated the increased intensities of both C−C (at 285 eV) and C = O bonds (at 288 eV), suggesting that the covalent bonds were majorly formed amongst CNFs. Moreover, the Ti 2p XPS spectra provided in Supplementary Fig. 10 reveal that the characteristic Ti−C bonds indicative of MXene integrity are preserved, as there is no evidence of new Ti−C bond formation. This suggests that the MXene nanosheets maintain their structural integrity throughout the aerogel fabrication process, even with the GA introduction. The XPS finding confirms that the intrinsic properties of the MXene nanosheets remained intact after the aerogel fabrication processes.

### Constructing a prediction model via active learning loops, data augmentation, and collaborative robots

Within the feasible parameter space, active learning loops and in silico data augmentation were employed to gather representative data points, and a high-accuracy prediction model for conductive aerogels was progressively constructed. During the active learning loops, two

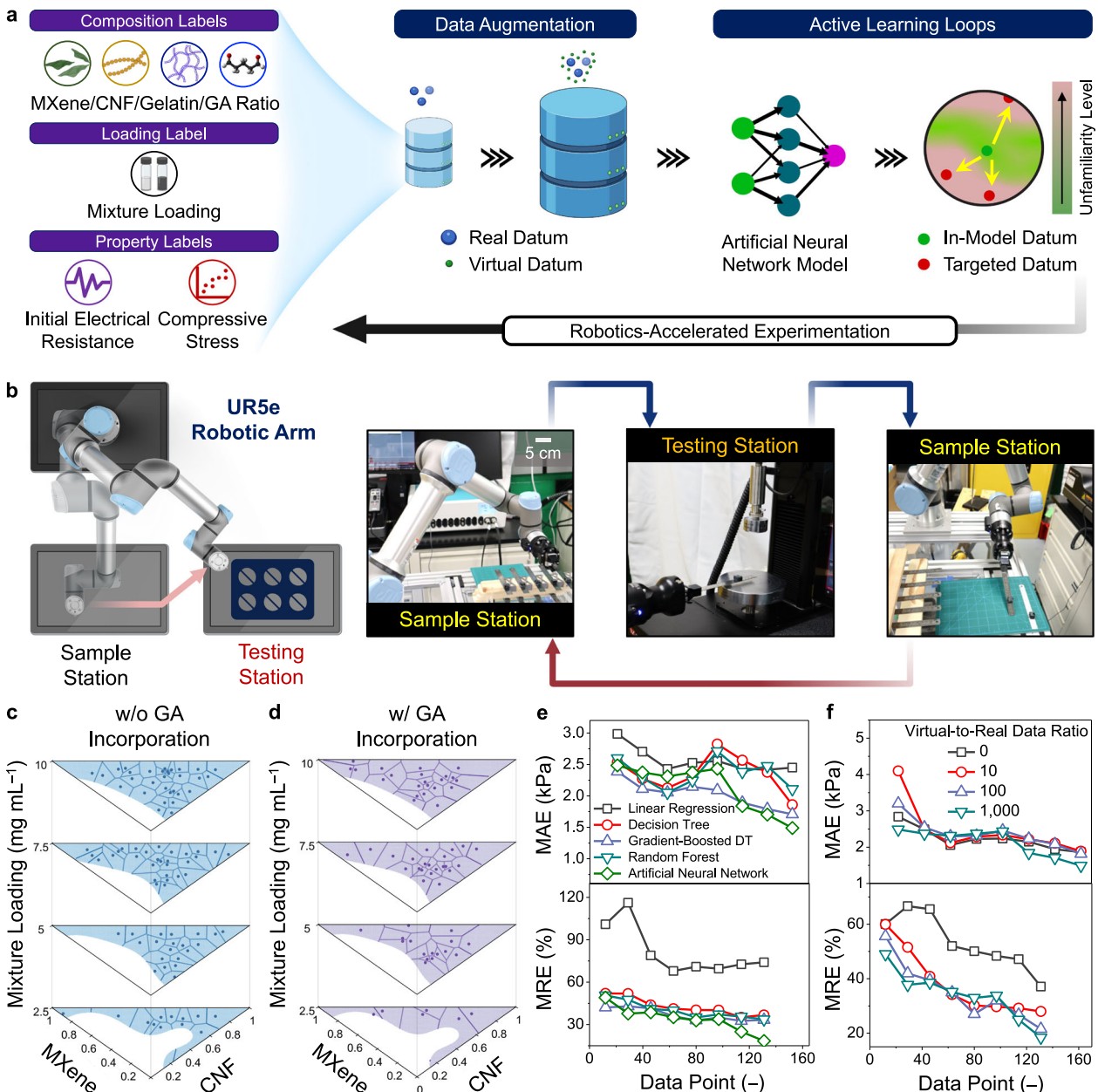

**Fig. 2 | Constructing a prediction model via active learning loops, data augmentation, and collaborative robots. a** Schematic illustration of a multi-stage AI/ML framework for constructing a prediction model via active learning loops, data augmentation, and robot-human teaming. **b** An autonomous testing platform integrated with a UR5e robotic arm and an Instron compression tester. 2D Voronoi tessellation diagrams (**c**) without and (**d**) with the GA incorporation after 8 active learning loops. **e** the mean absolute error (MAE),top, and the mean relative error (MRE), bottom, values of various prediction models based on linear regression, decision tree, gradient-boosted decision tree, random forest, and artificial neural network (ANN) algorithms. **f** MAE (top) and MRE (bottom) values of various ANN models based on different virtual-to-real data ratios.

collaborative robots, including an OT-2 robot and a UR5e-automated compression tester, were implemented to reduce the workload on human operators and enhance the data acquisition rates.

As illustrated in Fig. 2a, the active learning loops were initiated by commanding the OT-2 robot to prepare 20 aqueous mixtures at random MXene/CNF/gelatin/GA ratios and mixture loadings. Once vortexed, cast in silicone molds, and freeze-dried, the aqueous mixtures yielded the first batch of conductive aerogels. The MXene/CNF/gelatin/GA ratios of these conductive aerogels were recorded as the "composition" labels, while their mixture loadings served as the "loading" labels. Subsequently, the mechanical and electrical properties of these conductive aerogels were characterized. As

shown in Fig. 2b, to increase the data acquisition rates, a UR5e robotic arm was integrated with an Instron compression tester to achieve an autonomous testing platform. As demonstrated in Supplementary Movie 2, the UR5e arm was programmed to transfer conductive aerogels continuously from the sample station to the testing station. Once the UR5e arm completed placing a conductive aerogel at the testing station, an audio signal prompted the Instron tester to begin the compression test. After the test was finished, the Instron tester signaled the UR5e arm to remove the conductive aerogel and then position a new one. During the active learning loops, >400 conductive aerogels (3–4 replicates for each data point) were evaluated using the autonomous testing platform, and the

total operation time was estimated to be 81 h, averaging about 12 min to test one aerogel sample.

From the stress–strain curve of each conductive aerogel, the compressive stress at 30% strain (abbreviated as $\sigma_{30}$) was characterized, and the average $\sigma_{30}$ value from 3–4 aerogel replicates was designated as the "mechanical" label. Next, by using a two-electrode system (with a 1-cm gap between electrodes), the initial electrical resistance of each conductive aerogel (abbreviated as $R_0$) was measured, and the average $R_0$ value from 3–4 aerogel replicates was recorded as the "electrical" label. In short, each kind of conductive aerogel resulted in one data point, which included four "composition" labels, one "loading" label, one "mechanical" label, and one "electrical" label (see Supplementary Table 1). For one active learning loop, 20 new kinds of conductive aerogels were produced, therefore adding 20 data points to the database.

To improve model training efficiency and counteract potential overfitting, the User Input Principle (UIP) method was adopted to synthesize virtual data points (refer to Supplementary Note 6 for detailed description). The creation of virtual data points took place in the vicinity of collected real data points. For instance, Supplementary Fig. 11a, b demonstrate that, with slight variations in the MXene/CNF/gelatin/GA ratios (e.g., 64/24/12/+ vs. 62/26/12/+) led to the conductive aerogels with similar $\sigma_{30}$ (10.58 kPa vs. 10.63 kPa) and $R_0$ values (8.9 Ω vs. 10.1 Ω). Meanwhile, when the replicates of conductive aerogels were characterized, Supplementary Fig. 11c indicates that there were slight measurement variations in $\sigma_{30}$ and $R_0$. To synthesize virtual data points, Gaussian noises were introduced in the proximity of the composition, mechanical, and electrical labels based on our experimental observations. Afterward, both real and virtual data points were utilized as training data points for an ANN-based model using 4-fold cross-validation[70].

To collect more data points in the next active learning loop, the ANN model assessed the unfamiliarity level of each data point within the feasible parameter space using a hybrid acquisition function termed A Score, represented by Eq. (1)[71],

$$\text{A Score} = \hat{L} \cdot \hat{\sigma} \qquad (1)$$

where $\hat{L}$ denotes the Euclidean distance between in-model and model-targeted data points and $\hat{\sigma}$ denotes the ANN model's prediction variance (as detailed in Supplementary Note 7). The data points with the highest A Scores were the least familiar to the model and pinpointed for experimental validation in the next loop. By extracting the composition and loading labels of pinpointed data points, the OT-2 robot was activated to prepare a new set of MXene/CNF/gelatin/GA mixtures. Once vortexed, cast, and freeze-dried, a new batch of conductive aerogels was produced. Similarly, the $\sigma_{30}$ and $R_0$ values of these conductive aerogels were characterized via the autonomous testing platform and the two-electrode system, respectively. Based on these real data points, virtual data points were synthesized using the UIP method. Upon inputting the real and virtual data points, the ANN model was retrained, re-assessed A Scores, and suggested another set of fabrication parameters for the next active learning cycle.

With the operation of two collaborative robots, the active learning loops were largely facilitated. Each loop took an average of 2.5 days: 2 h dedicated to OT-2 pipetting, 48 h for freeze-drying, 4 h allocated to autonomous testing, and another 4 h for model training. In total, 8 active learning loops were carried out, and 162 kinds of conductive aerogels were stagewise produced (refer to Supplementary Table 4). In this work, we collected 162 real data points during 8 active learning loops. Afterwards, to improve model training efficiency and counteract potential overfitting, the UIP method was adopted to synthesize virtual data points in a 1-to-1000 ratio, thus leading to the generation of ~160,000 virtual data points. Afterward, both real and virtual data

points were utilized as training data points for constructing the ANN model using 4-fold cross-validation.

During the active learning loops, the ANN model continued to evolve and was evaluated from two perspectives: (1) the distribution of collected data points and (2) the accuracy of multi-property predictions. First, as shown in Fig. 2c, d and Supplementary Figs. 12, 2D Voronoi tessellation diagrams were plotted to visualize how data points were sequentially collected and distributed within the feasible parameter space. During active learning loops, the ANN model efficiently explored the feasible parameter space and guided experiments towards the unfamiliar regions, effectively mitigating the rise of redundant data clusters.

Second, the accuracy of multi-property predictions was assessed using a set of testing data points, which were never input into the model (see Supplementary Table 5). For each testing data point, the ANN model provided the predicted $\sigma_{30}$ and $R_0$ values based on the "composition" and "loading" labels. Then, the model-predicted $\sigma_{30}$ and $R_0$ values were compared with the actual $\sigma_{30}$ and $R_0$ values of the testing data point. The deviation between model-predicted and actual $\sigma_{30}$ values was evaluated using the mean absolute error (MAE), as detailed in Eq. (2),

$$\text{MAE} = \frac{1}{N} \sum_{i=1}^{N} \left| \text{predicted } \sigma_{30}^i - \sigma_{30}^i \right| \qquad (2)$$

where $N$ is the cumulative number of testing data points, predicted $\sigma_{30}^i$ is the model-predicted $\sigma_{30}$ value based on a testing data point ($i$), $\sigma_{30}^i$ is the actual $\sigma_{30}$ value of a testing data point ($i$). On the other hand, the deviation between model-predicted and actual $R_0$ values were assessed using the mean relative error (MRE), as detailed in Eq. (3),

$$\text{MRE} = \frac{1}{N} \sum_{i=1}^{N} \left| \frac{\log(\text{predicted } R_0^i) - \log(R_0^i)}{\log(R_0^i)} \right| \qquad (3)$$

where $N$ is the cumulative number of testing data points, predicted $R_0^i$ is the model-predicted $R_0$ value based on a testing data point ($i$), $R_0^i$ is the actual $R_0$ value of a testing data point ($i$). Smaller MAE and MRE values indicated higher prediction accuracy, while larger values indicated lower accuracy. By evaluating MAEs and MREs, we were able to assess the model's prediction accuracy in predicting the mechanical and electrical properties of conductive aerogels from their fabrication parameters.

Throughout 8 active learning loops, the MAE (that assessed the accuracy of $\sigma_{30}$ prediction) continually decreased from 2.5 to 1.5 kPa (Fig. 2e, top), and the MRE (that assessed the accuracy of $R_0$ prediction) decreased from 49.0% to 18.4% (Fig. 2e, bottom). Towards the end of active learning loops, both MAE and MRE values were stabilized and gradually approached toward the measurement variations of $\sigma_{30}$ (-1.1 kPa) and $R_0$ (-10.1%). Among other models based on linear regression, decision tree, gradient-boosted decision tree, random forest algorithms, the ANN model demonstrated the lowest MAE and MRE values (Fig. 2e). Furthermore, without conducting data augmentation, the ANN model presented a higher MAE of >1.9 kPa (for $\sigma_{30}$ prediction) and a higher MRE of >37% (for $R_0$ prediction), due to the model overfitting upon the use of a small database (Fig. 2f). As the virtual-to-real data ratio increased to 100 and 1000, the MAE values decreased to 1.8 kPa and 1.5 kPa, and the MRE values decreased to 21.5% and 18.4%, respectively. In this work, the optimal virtual-to-real data ratio was set to be 1000, which enabled high learning efficiency and still kept the model training time below 4 h. On the other hand, when the virtual-to-real data ratio increased to 5000 and 10,000, the model training time increased to >1 and >2 days, respectively. Finally, the ANN model that demonstrated the lowest MAE and MRE values was selected as "the champion model", which was deployed the next to

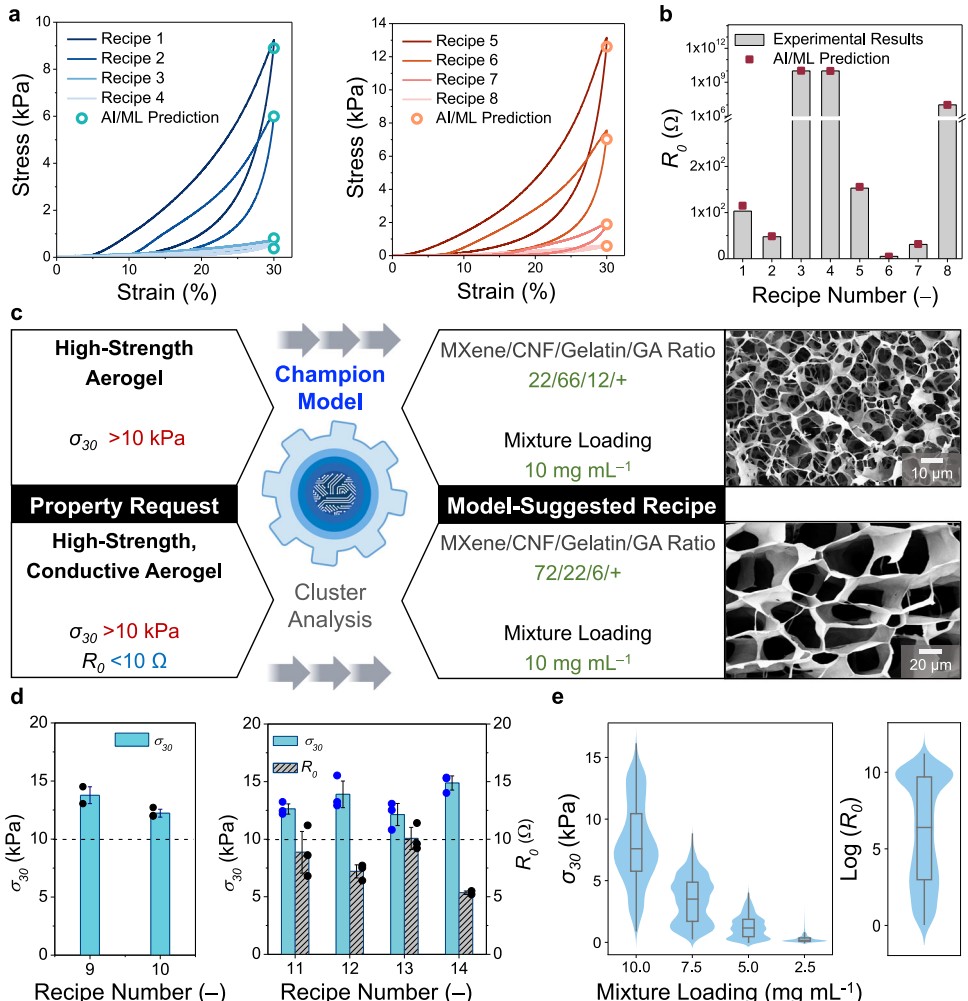

**Fig. 3 | Predicting compressive strengths and electrical resistances of conductive aerogels. a** Comparison between the actual stress–strain curves of conductive aerogels (recipes #1–#8) and the model-predicted $\sigma_{30}$ values. **b** Comparison between the actual initial electrical resistances of conductive aerogels (recipes #1–#8) and model-predicted $R_0$ values. **c** By inputting specific design requests, the champion model was able to automate the inverse design processes of conductive aerogels by directly suggesting suitable sets of fabrication parameters, without the need for iterative optimization experiments. Inset shows the SEM images of two model-suggested conductive aerogels. **d** Comparison between actual and model-predicted $\sigma_{30}$ (left) and $R_0$ (right) values of conductive aerogels (recipes #9–#14). Data are presented as mean ± s.d., $n = 3$, with each independent experiment marked by a black or blue dot. **e** Violin plots of achievable $\sigma_{30}$ and $R_0$ values of conductive aerogels. The embedded box plot within each violin plot indicates the 25th and 75th percentiles with the median represented by the center line. Whiskers extend to 1.5 ×IQR from the box, $n = 491,131$. Error bars represent s.d.

automate the design of conductive aerogels with programmable mechanical and electrical properties.

In addition, the selection of a suitable sampling method is important for the construction of an accurate prediction model. Comparative analyses of different sampling methods were conducted: random sampling, Latin hypercube sampling, and active learning sampling (as demonstrated in our study). Supplementary Fig. 13 and Supplementary Table 6 provide details on the three sampling cycles for each method, with five separate physical experiments carried out in each cycle. As shown in Supplementary Fig. 14a, the active learning sampling method outperformed the others, achieving a success rate of >95% in recommending the MXene/CNF/gelatin/GA ratios that resulted in the production of A-grade aerogels. In comparison, random sampling and Latin hypercube sampling yielded lower success rates of 80% and 67%, respectively. After the three sampling cycles, we applied the UIP method to the real data points collected from the different sampling methods and synthesized virtual data points at a ratio of 1-to-1000. Using these real and virtual data points, we trained multiple ANN-based prediction models. As shown in Supplementary Fig. 14b,

among all the sampling methods, the prediction model trained on data points from the active learning sampling exhibited superior learning efficacy and enhanced prediction accuracy, achieving the lowest recorded MAE values of 1.1 kPa. Whereas the MAEs from random sampling and Latin hypercube sampling were 5.8 kPa and 5.0 kPa, respectively, after completing three cycles.

### Predicting compressive strengths and electrical resistances of conductive aerogels

By leveraging the champion model's prediction capabilities, two-way design tasks were successfully demonstrated, involving (1) predicting the $\sigma_{30}$ and $R_0$ values of a conductive aerogel from its fabrication parameters and (2) suggesting an ideal set of MXene/CNF/gelatin/GA ratio and mixture loading to produce a conductive aerogel with user-designated characteristics.

First, by selecting different sets of fabrication parameters, various conductive aerogels were fabricated and characterized (recipes #1–#8 in Supplementary Table 7). As evidenced in Fig. 3a, b, the champion model predicted the $\sigma_{30}$ and $R_0$ values of these conductive aerogels

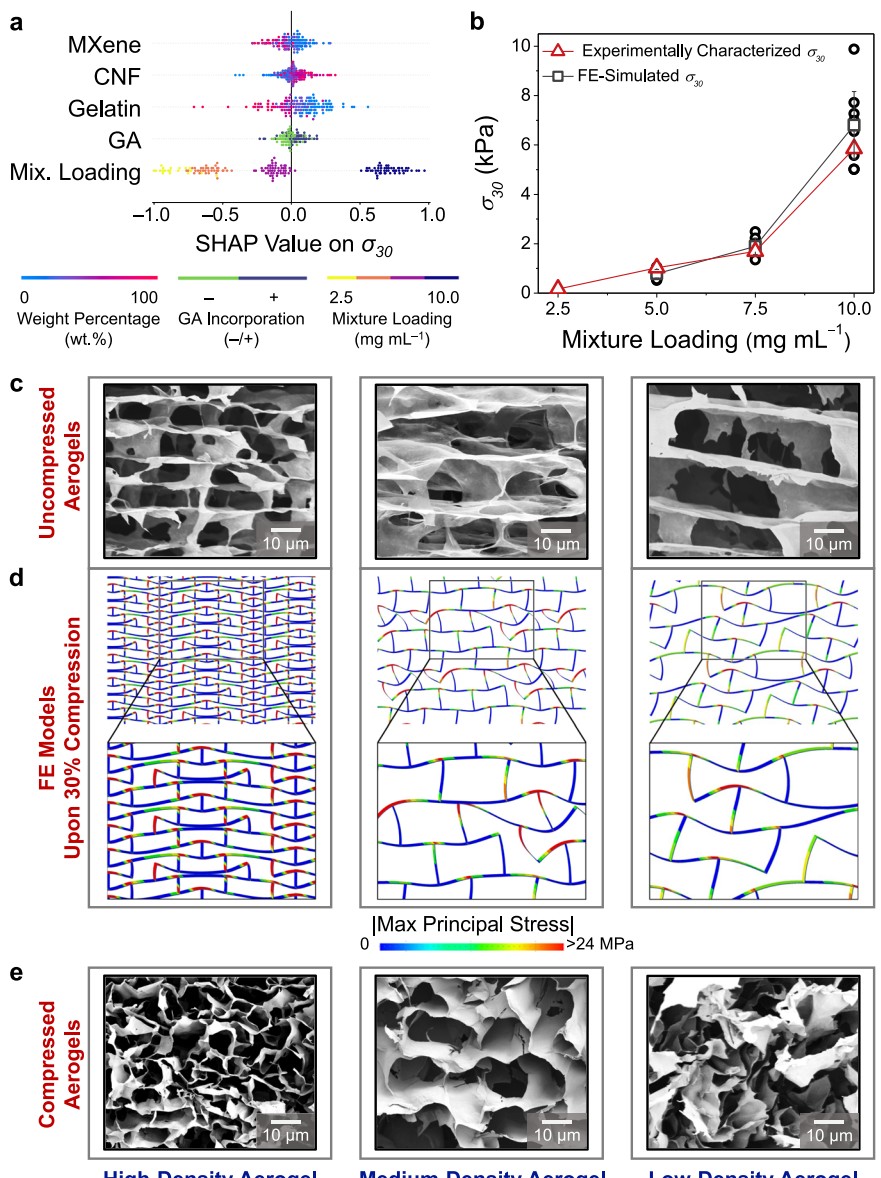

**Fig. 4 | SHapley Additive exPlanations (SHAP) model interpretation and Finite Element (FE) simulations to uncover complex fabrication–structure–property correlations.** (**a**) Normalized SHAP values of MXene, CNF, gelatin, GA loadings, and mixture loading on the $\sigma_{30}$ values of conductive aerogels. **b** Comparison between the FE-simulated and experimentally characterized $\sigma_{30}$ values of conductive aerogels at the same MXene/CNF/gelatin/GA ratio yet at different mixture loadings. Data are presented as mean ± s.d., $n = 3$, with each independent experiment marked by an open black circle. **c** SEM images of high-, medium-, and low-density conductive aerogels at their uncompressed states. **d** Localized stress distribution profiles of high-, medium-, and low-density aerogel models (from FE simulations) under 30% compression. **e** SEM images of high-, medium-, and low-density conductive aerogels at their compressed states. Error bars represent s.d.

accurately based on their "composition" and "loading" labels, and the predicted values were close to the experimentally characterized results. Second, the inverse design of conductive aerogels was automated by the champion model, without the need for iterative optimization experiments. As illustrated in Fig. 3c, two conductive aerogels were targeted with specific property requirements, including (1) high-strength aerogels ($\sigma_{30} > 10$ kPa) and (2) high-strength, conductive aerogels ($\sigma_{30} > 10$ kPa and $R_0 < 10$ Ω). By inputting these design requests, the champion model performed clustering analyses to pinpoint the most suitable sets of fabrication parameters. By following the model-suggested fabrication parameters, two kinds of conductive aerogels were produced. As demonstrated in Fig. 3d, for the design request #1, the champion model suggested two aerogels (recipes #9–#10 in Supplementary Table 8), both of which demonstrated the $\sigma_{30}$ values that were higher than the input requirement of 10 kPa. For

the design request #2, the champion model suggested four aerogels (recipes #11–#14 in Supplementary Table 8), showing the average $\sigma_{30}$ and $R_0$ values > 10 kPa and <10 Ω, respectively. The inset of Fig. 3c and Supplementary Fig. 15 show the SEM images of the model-suggested aerogels (recipes #9 and #14). As displayed in the violin plots (Fig. 3e), the achievable $\sigma_{30}$ and $R_0$ values of conductive aerogels spanned widely between $0.1 < \sigma_{30} < 16.0$ kPa and $10^0 < R_0 < 10^{10}$ Ω, through navigating the DOFs of MXene/CNF/gelatin/GA ratios and mixture loadings.

## SHAP model interpretation and FE simulations to uncover complex fabrication–structure–property correlations
To address the "black box" nature of the champion model, the SHAP model interpretation method was applied to 162 data points collected during active learning loops. SHAP relies on a game theoretic approach

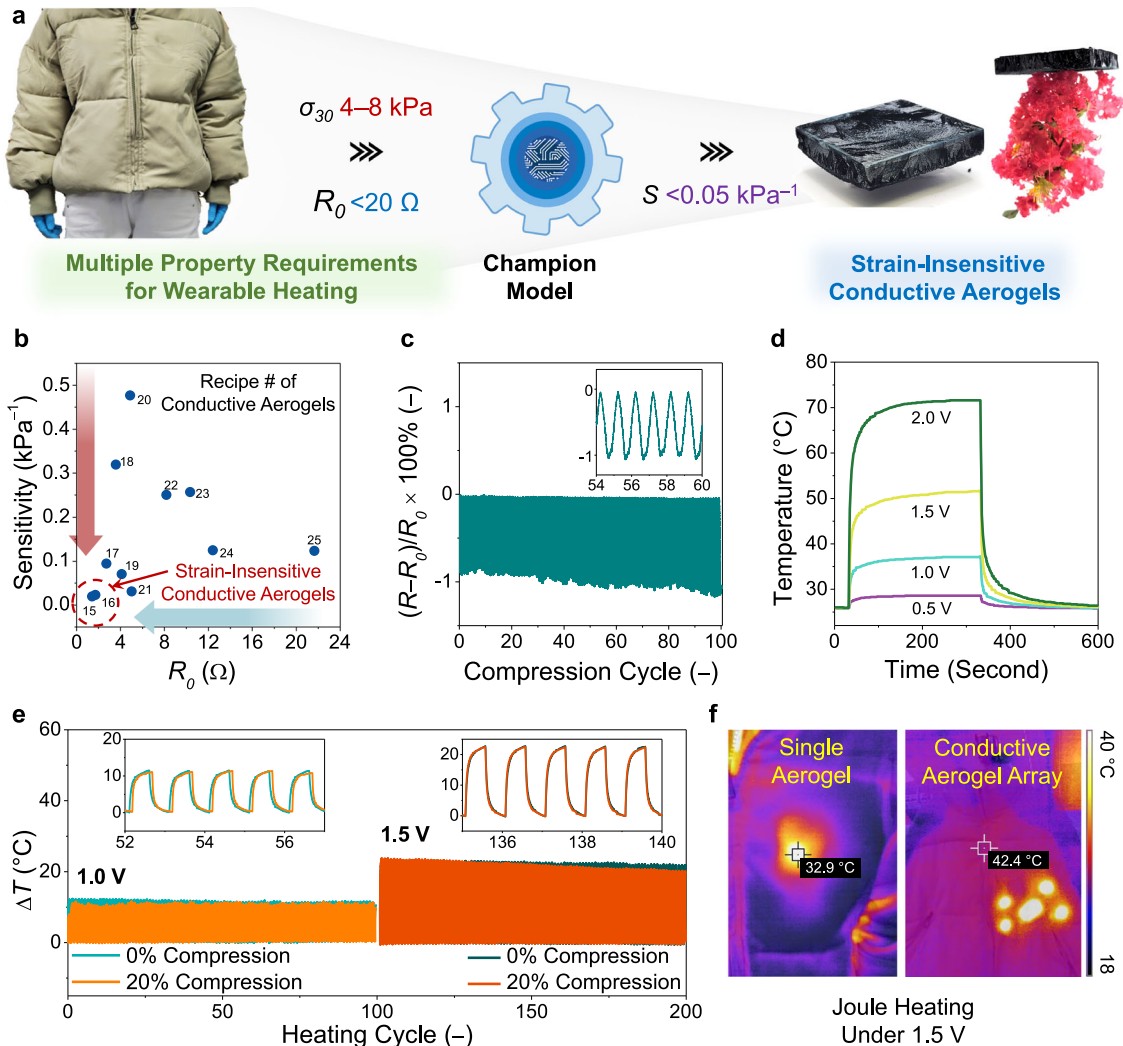

**Fig. 5 | Machine intelligence accelerated discovery of strain-insensitive conductive aerogels for wearable thermal management. a** Schematic illustration of machine intelligence design process of strain-insensitive conductive aerogels. **b** $R_0$–sensitivity profile of model-suggested conductive aerogels. **c** Time-resolved relative resistance changes of a strain-insensitive conductive aerogel under 100 cycles of 20% compression. **d** Temperature–time profiles of a strain-insensitive conductive aerogel at different applied voltages. **e** Time-resolved temperature profiles of a strain-insensitive conductive aerogel at its relaxed and 20% compressed states under Joule heating (at 1.0 and 1.5 V). **f** Thermal images of the aerogel-incorporated jacket under Joule heating at 1.5 V.

and contains a permutation explainer program to explain the output of any AI/ML model[72]. By iterating over complete permutations of the features, the SHAP values are calculated to approximate the contribution of each fabrication parameter to a specific property. A positive SHAP value indicates a positive correlation, and vice versa (see Supplementary Note 8 and Supplementary Fig. 16 for further details). In this study, the SHAP values of MXene, CNF, gelatin, GA loadings, and mixture loading on the $\sigma_{30}$ and $R_0$ values of conductive aerogels were calculated. Figure 4a shows that the SHAP value of mixture loading on $\sigma_{30}$ ranged the widest from −1.00 to +0.97 among the others (e.g., MXene loading from −0.28 to +0.28, CNF loading from −0.41 to +0.32, gelatin loading from −0.71 to +0.56, and GA loading from −0.23 to +0.19). The above SHAP results suggest that the mixture loading of aqueous mixture (that affected the density of a conductive aerogel) presented the most significant impact on the mechanical properties.

On the other hand, Supplementary Fig. 17 shows that the SHAP values for MXene loading on $R_0$ ranged from −0.94 to +0.97, demonstrating the most substantial variability and pronounced impact compared to other factors: CNF loading (−0.41 to +0.65), gelatin loading (−0.83 to +0.94), GA loading (−0.24 to +0.36), and mixture loading (−0.35 to +1.00). These SHAP results highlight that

increasing the MXene loading significantly lowered the $R_0$ of conductive aerogels, marking it as the most influential factor. As detailed in Supplementary Fig. 18, the mixture loading notably correlated positively with the aerogel density. Similarly, increasing the mixture loading, which corresponds to increased aerogel density, had a similar effect on lowering the $R_0$ of conductive aerogels. However, the influence of mixture loading (or aerogel density) was statistically less significant than that of MXene loading, and it showed a high degree of correlation with the MXene/CNF/gelatin/GA ratio.

To validate the SHAP-identified correlations, additional experiments were conducted in Supplementary Fig. 19. With the MXene/CNF/gelatin/GA ratio fixed at 11/77/12/+, conductive aerogels consistently exhibited high $R_0$ values with minimal variations across different mixture loadings (ranging from 10.0 to 2.5 mg mL$^{-1}$) as demonstrated in Supplementary Fig. 19a. The corresponding SEM images are provided in Supplementary Fig. 19b. Conversely, when the MXene/CNF/gelatin/GA ratio was set to 61/28/11/−, conductive aerogels displayed a significant decrease in their $R_0$ values but followed a similar trend across different mixture loadings with values ranging from 2.1 to 17.3 Ω (Supplementary Fig. 19a). The

corresponding SEM images are provided in Supplementary Fig. 19c. In contrast, the conductive aerogels prepared from the MXene/CNF/gelatin/GA ratio of 43/42/15/–, showed pronounced variations in $R_0$ values spanning from 7.5 to 228.4 $\Omega$ at different mixture loadings (Supplementary Fig. 19a). The corresponding SEM images are provided in Supplementary Fig. 19d. These results show that the MXene loading exhibited a higher impact on the $R_0$ of conductive aerogels than the mixture loading.

Next, to investigate strong correlations between mixture loading and $\sigma_{30}$ value, four conductive aerogels were fabricated at the same MXene/CNF/gelatin/GA ratio of 80/20/0/– but at different mixture loadings (from 2.5 to 10.0 mg mL$^{-1}$). As shown in Fig. 4b, as the mixture loadings increased, the $\sigma_{30}$ values of conductive aerogels rose significantly from 0.2 kPa to 5.9 kPa. Four other conductive aerogels were produced at the same mixture loading of 10.0 mg mL$^{-1}$ but with various MXene/CNF/gelatin/GA ratios (from 80/20/0/– to 20/80/0/–). Despite the difference in aerogel composition, the $\sigma_{30}$ values exhibited only a minor shift to 7.5 kPa from 5.9 kPa. Supported by SHAP analyses and experimental results, it was determined that adjusting the mixture loading level was a more effective method for tuning the compression resilience of a conductive aerogel (e.g., $\sigma_{30}$).

To delve deeper into the mechanistic effects of mixture loadings on the mechanical properties of conductive aerogels, three FE models were constructed. These FE models represented the conductive aerogels at the same MXene/CNF/gelatin/GA ratio (80.0/20.0/0.0/–) yet at different mixture loadings (10.0, 7.5, 5.0 mg mL$^{-1}$). These FE models were named as high-, medium-, and low-density aerogel models. Extracted from the SEM images in Fig. 4c and summarized in Supplementary Table 9, several structural features of conductive aerogels, such as pore dimensions, wall thicknesses, and fracture densities, were input to construct these FE models using a commercial package of ABAQUS/CAE 2020. In the FE simulations, the microstructures of conductive aerogels were represented as the 4 × 4 supercells containing two-dimensional staggered lattices using second-order plane strain elements, and the periodic boundary conditions were imposed. Then, wall discontinuities were randomly introduced into the FE models to account for the structural imperfections of conductive aerogels. Supplementary Note 9, Supplementary Fig. 20, and 21 provide further details regarding the construction of FE models. Next, by exerting a vertical compressive strain of 30%, the $\sigma_{30}$ values of three FE models were simulated in Fig. 4b, showing good agreement with the experimentally characterized $\sigma_{30}$ values. Figure 4d showcases the FE models of conductive aerogels under 30% compression to provide insights regarding internal displacements and localized stress distributions. As compared in Fig. 4e, the FE models under 30% compression cohered with the SEM observations of conductive aerogels at their compressed states.

The high-density aerogel exhibited small, closely packed pores (with the average size of 18.2 × 6.4 $\mu m^2$) and thus allowed for uniform stress distributions upon compression, largely suppressing internal displacements of compartments. In the high-density aerogel, the localized stress tended to concentrate at the corners of each compartment, while the MXene-based walls were robust enough (with Young's modulus at 3.4 GPa) to prevent significant deformation toward cracking. On the other hand, as the mixture loading decreased, the medium- and low-density aerogels had larger and wider pores (28.3 × 17.8 and 39.1 × 19.0 $\mu m^2$), which were less efficient to transmit vertical stresses to neighboring compartments. As a result, the pores were likely to deform and distort upon the localized stress, resulting in significant internal displacements of compartments. Through the combined use of SHAP analyses, experimental validation, and FE simulations, we provided a promising solution to the "black box" challenges often associated with AI/ML predictions, enhancing the champion model's interpretability.

## Machine intelligence accelerated discovery of strain-insensitive conductive aerogels for wearable thermal management

Conductive aerogels offer promising applications in personal thermal management, owing to their lightweight feature, high electrical conductivity, and thermal insulation properties[14]. Fig. 5a outlines the machine intelligence accelerated design process to fabricate a strain-insensitive conductive aerogel suitable for wearable heating applications. First, two property requirements were considered: (1) compatible compressive strength with current filling materials ($\sigma_{30}$ ~ 4–8 kPa) and (2) high electrical conductivity for efficient Joule heating ($R_0 < 20\,\Omega$). Upon inputting these design requests, the champion model was able to suggest multiple sets of fabrication parameters, and various conductive aerogels that met two property requirements were fabricated (see Supplementary Table 10).

Next, an additional criterion of low pressure sensitivity was set to discover a strain-insensitive conductive aerogel, ensuring strain-stable Joule heating performance under repetitive compression. The definition of pressure sensitivity is provided in Eq. (4),

$$\text{Pressure Sensitivity}\,(S) = \frac{\left| (R - R_0)/R_0 \right|}{\sigma_{30}} \qquad (4)$$

As demonstrated in the $R_0$–sensitivity profiles (Fig. 5b), the conductive aerogel based on recipe #16 was selected (at the MXene/CNF/gelatin/GA ratio of 78/13/9/– and at the mixture loading of 7.5 mg mL$^{-1}$). The model-suggested conductive aerogel exhibited a $\sigma_{30}$ value of 4.0 kPa, a low $R_0$ value of 1.7, and an ultralow pressure sensitivity of 0.02 kPa$^{-1}$. Figure 5c demonstrates that the relative resistance changes of the strain-insensitive conductive aerogel was only 0.9% under 100 cycles of 20% compression. Next, the Joule heating performance of the strain-insensitive conductive aerogel was investigated by applying various voltages, as evidenced in the measured temperature–time profiles (Fig. 5d). At the applied voltages of 0.5, 1.0, 1.5, and 2.0 V, the strain-insensitive conductive aerogel demonstrated sharp temperature increases up to 29, 37, 51, and 70 °C, respectively, within 300 s. As shown in Supplementary Fig. 22, the strain-insensitive conductive aerogel displayed a linear relationship between the maximum temperature and the square of the applied voltage, well adhering to Joule's law.

To further assess the Joule heating performance under compression, the strain-insensitive conductive aerogel at both relaxed and 20% compressed states was subject to 100 heating cycles at 1.0 and 1.5 V. As shown in Fig. 5e, the conductive aerogels demonstrated strain-unresponsive heating/cooling profiles, with stable average temperature variations for 100 heating cycles. Such efficient and strain-stable Joule heating performance was well-suited for wearable heating applications. Additionally, we conducted thermal conductivity measurements to assess the thermal insulation properties of the strain-insensitive conductive aerogel. The density and thermal insulation performance of the strain-insensitive conductive aerogel were characterized as 10.1 mg cm$^{-3}$ and 0.034 W mK$^{-1}$, respectively. Compared with other thermal insulation materials in Supplementary Fig. 23, the strain-insensitive conductive aerogel offered lightweight features, competitive thermal insulation properties, and efficient Joule heating performance. To demonstrate its practical exploitation, multiple strain-insensitive conductive aerogels with the dimensions of 5.0 × 5.0 × 1.0 cm$^3$ were inserted into a commercial jacket. Powered by a portable battery system (at 1.5 V), the aerogel-incorporated jacket was heated on demand to warm the subregion of a volunteer (thermal images in Fig. 5f).

Compared with the state-of-the-art works of conductive aerogels, our robotics/ML-integrated workflow demonstrates multiple advancements, as summarized in Supplementary Table 11[73–75], 12[2,8,19,20,29,33,76–85], and 13[8,19,20,29,33,76–82]. The first advancement is to construct a prediction model with high accuracy across the entire

parameter space. As detailed in Supplementary Table 11, the state-of-the-art works typically focus on pinpointing an optimal set of fabrication parameters to maximize device performance or identifying precise conditions necessary for successful chemical reactions. While some of these works also utilize high-throughput robotic platforms to navigate a parameter space with multiple degrees of freedom (DOFs), they do not intend to examine data points that stray from the primary objectives, such as determining the most effective parameter combination or sequence of reaction steps. Consequently, if the design requirements shift—for instance, the synthesis of a new chemical compound—new round(s) of optimization experiments would be necessary. This requirement for additional experimentation may not align well with demands for rapid customization. In contrast to traditional methods, our integrated workflow that combines collaborative robotics with SVM classification, active learning loops, and data augmentation, resulted in a prediction model that consistently delivers high accuracy across the entire parameter space. Consequently, to respond to diverse design requests for customization, our prediction model can automatically identify the optimal fabrication parameters, eliminating the need for repeated optimization experiments.

The second advancement is to automate the design of conductive aerogels with programmable properties. Through the robotics/ML-integrated workflow, the prediction model can perform two-way design tasks: (1) predicting the physicochemical properties of conductive aerogels based on their fabrication parameters and (2) automating the inverse design of conductive aerogels to align with specific property requirements. The prediction model is well-suited for crafting conductive aerogels with customizable mechanical and electrical properties. Given the laborious nature of self-assembly and the absence of robotic systems available for fully automating freeze drying processes, our prediction model can directly identify viable and optimal fabrication parameters, without the need for iterative optimization experiments. Compared with the state-of-the-art works of conductive aerogels (see Supplementary Table 12), our automatic design method demonstrates superior efficiency in discovering conductive aerogels with programmable properties.

The third advancement is to elucidate data-driven design principles in complex material systems. Supplementary Table 13 presents a comparison of our work with multiple studies focusing on MXene-based aerogels, including MXene/CNT, MXene/rGO, MXene/ANF, and MXene/CNF. Traditionally, these state-of-the-art works explore the intricate correlations between fabrication parameters and aerogel properties through the one-factor-at-a-time (OFAT) design of experiment (DoE) method. The OFAT DoE method has yielded diverse design insights across different MXene-based aerogel systems. For example, Zeng et al., Xu et al., Jiang et al., and Wang et al. groups discovered that increasing the MXene loading decreased the compressive strength and electrical resistance of conductive aerogels[33,76,80,81]. In contrast, our research harnesses the synergy of collaborative robotics and advanced AI/ML predictive analytics. By constructing a robust prediction model followed by model interpretation (e.g., SHAP), we can investigate several component–property correlations and generalize design principles with statistical significance, offering a more systematic understanding of complex material systems.

## Discussion

In conclusion, an integrated workflow, which combined automated robotic experiments with an AI/ML framework, consisted of four main phases: screening, analysis, design/optimization, and self-correction/validation. In the screening phase, we used an automated pipetting robot (i.e., OT-2) to efficiently prepare 264 different mixtures with varying ratios of MXene/CNF/gelatin/GA and mixture loadings, replacing laborious manual experimentation. After the freeze drying process, we assessed the structural integrity of conductive aerogels to train a SVM classifier, which played a critical role in defining a feasible parameter space with high DOFs. In the analysis phase, we avoided the traditional OFAT method and instead employed a more robust strategy involving 8 active learning cycles and data augmentation. During active learning, we fabricated and characterized 162 different types of conductive aerogels, uncovering intricate correlations between fabrication parameters and aerogel properties. Afterward, we used a data augmentation technique to enrich our dataset, enabling the construction of an ANN-based prediction model with high accuracy. In the design/optimization phase, we leveraged the prediction model to perform two-way design tasks: (1) predicting the physicochemical properties of conductive aerogels based on fabrication parameters, and (2) automating the inverse design of conductive aerogels to meet specific property criteria. In the self-correction/validation phase, our prediction model surpassed traditional experience-based design approaches by conducting clustering analyses to identify optimal fabrication parameters that aligned with targeted design requests. Furthermore, we used model interpretation and FE simulations to validate the established correlations. Finally, the model's predictive power was harnessed to discover a strain-insensitive conductive aerogel with compatible mechanical strength, high electrical conductivity, and ultralow pressure sensitivity suitable for wearable heating applications. The successful integration of robotic experiments, AI/ML predictions, and FE simulations has demonstrated a synergistic approach, with potential applications beyond conductive aerogels, such as tactile sensors[86,87], stretchable conductors[88,89], catalysts[59,60], and electrochemical electrolyte optimization[57,58].

Meanwhile, we would like to emphasize that the integration of collaborative robotics and AI/ML predictions into the research workflow aims to enhance, rather than replace, the expertise of researchers. While our robotics/ML-integrated system aims to reduce the experimental burden and improve efficiency, the domain knowledge and experience of materials researchers remain crucial in collecting high-quality data points and constructing a prediction model. In our work, the expertise of researchers was indispensable at several pivotal points. First, after using the OT-2 robot to prepare a library of conductive aerogels, it was imperative for researchers to grade each aerogel based on its structural integrity. This critical evaluation largely relied on the judgment of experienced researchers and ensured a robust screening process. Subsequently, when establishing the SVM model, the determination of the A-grade probability threshold was another point where researchers' insights were important. Setting this threshold was a balancing act. If the threshold was set too high, the design space would become constricted, potentially inhibiting the innovation of high-performance conductive aerogels. If the threshold was set too low, it would likely result in an unfavorable rate of experimental failure, impeding the efficiency of data collection in active learning cycles. Moreover, the data curation phase was essential for the development of an accurate predictive model. As evidenced in Supplementary Fig. 7, incorporating complete stress–strain curves of conductive aerogels into the training dataset led to a notable decrease in prediction accuracy, as reflected by the MAE value increased to 4.5 kPa. The selection of suitable property labels for model training was therefore contingent upon the researchers' deep understanding and domain expertise. Finally, the data augmentation phase was informed by researchers' empirical observations, as depicted in Supplementary Fig. 11. These observations were foundational for the generation of virtual data points, which were enhanced with Gaussian noise to simulate real-world variability. Through each of these phases, the contribution of researchers was crucial, not only in applying their experiential knowledge for qualitative assessments but also in exercising their domain expertise to guide the quantitative modeling and data analysis.

## Methods

### Materials

Lithium fluoride (LiF, BioUltra, ≥99.0%), hydrochloric acid (HCl, 37%), northern bleached softwood kraft (NBSK) pulp (NIST® RM 8495), TEMPO (99%), sodium bromide (NaBr, ≥99.5%), sodium bicarbonate ($NaHCO_3$, ≥99.7%), sodium hydroxide (NaOH, ≥98%), and gelatin (from porcine skin, Type A) were purchased from Sigma-Aldrich. $Ti_3AlC_2$ MAX powders were purchased from Lai Zhou Kai Kai, China. Silicone molds were purchased from Amazon. Deionized water (18.2 MΩ) was obtained from a Milli-Q water purification system (Millipore Corp., Bedford, MA, USA) and used as the water source in this work.

### Preparation of $Ti_3C_2T_x$ MXene nanosheets

$Ti_3C_2T_x$ MXene nanosheets were prepared according to the literature[90]. First, 3.0 g of LiF was dissolved in 6.0 M HCl at 35 °C under vigorous stirring. After the dissolution of LiF, 1.0 g of $Ti_3AlC_2$ MAX powder was slowly added into the HF-containing solution. The mixture was kept at 35 °C for 24 h. Afterward, the solid residue was washed with 2.0 M HCl and DI water sequentially until the pH value increased to 6. Subsequently, the washed residue was added into 100 mL of DI water, ultrasonicated for 1 h, and centrifuged at 5000 × g. for 30 min. The supernatant was collected as the final suspension of MXene nanosheets with the concentration of 10–12 mg mL$^{-1}$.

### Preparation of CNF dispersion

The CNF dispersion was prepared according to the literature[91]. First, 20 g of NBSK pulp was suspended in 1.0 L of DI water, and then TEMPO ($2 \times 10^{-3}$ mole) and NaBr (0.02 mole) were added into the pulp. The TEMPO-mediated oxidation was initiated by adding 0.2 mole of NaClO, and the oxidation process was maintained for 5–6 h under continuous stirring, during which the pH was controlled at 10.0 by adding NaOH solution. The TEMPO-oxidized pulp was repeatedly washed with DI water until the pH returned back to 7.0. Afterward, the pulp was disassembled in a microfluidizer processor (Microfluidics M-110EH), and the concentration of CNF dispersion was about 10 mg mL$^{-1}$.

### Preparation of gelatin solution

10.0 g of gelatin was dissolved in 1.0 L of DI water followed by continuous stirring for 48 h, and the concentration of gelatin solution was 10.0 mg mL$^{-1}$.

### Robot-assisted fabrication of conductive aerogels

An automated pipetting robot (OT-2, Opentrons) was operated to prepare different mixtures with varying MXene/CNF/gelatin ratios. For each mixture, the dispersions of MXene nanosheets and CNFs as well as the gelatin solution were mixed at different volumes. The mixture loadings of MXene/CNF/gelatin mixtures were controlled to be 10.0, 7.5, 5.0, and 2.5 mg mL$^{-1}$. The solution containing 25 wt.% of GA was optionally added into the MXene/CNF/gelatin mixtures at a consistent ratio of 35 μL for every 100 mg of solid content. Afterward, the robot-prepared MXene/CNF/gelatin/GA mixtures were vortexed at 2000 r.p.m. for 30 s and poured into silicone molds. Silicone molds with cavities of 15 × 15 mm and a height of 12 mm were used to prepare cuboid aerogel samples. The slurry-like mixtures were refrigerated overnight at 4.0 °C and then frozen using liquid nitrogen followed by a freeze-drying process (at −85 °C and 10$^{-3}$ atm, Labconco FreeZone), allowing for the production of conductive aerogels.

### Compression tests of conductive aerogels

The stress–strain curves of conductive aerogels (with average dimensions of 1.5 × 1.5 × 1.0 cm$^3$) were measured by a mechanical testing machine (Instron 68SC-05) with a 500 N load cell for 5 cycles of loading and unloading (at 5 mm s$^{-1}$ for 30% compression). A collaborative

robotic arm (UR5e, Universal Robots) was integrated into a universal testing system to automate the compression tests.

### Electrical resistance measurements of conductive aerogels

The initial electrical resistance, $R_0$, of the conductive aerogels were measured using a two-electrode system connected with an electrochemical workstation (Autolab PGSTAT302N, Metrohm) or an Industrial Multimeter (Fluke 87 V). The gap between two electrodes was fixed at 1 cm. A sample subjected to electrical resistance testing was not employed again for compression testing, and vice versa. It ensured that each test was conducted on an uncompromised conductive aerogel, preserving the integrity of our results.

### Pressure sensitivity measurements of conductive aerogels

The conductive aerogels were initially set up on a two-electrode system and linked to an electrochemical workstation (Autolab PGSTAT302N, Metrohm). To determine the pressure sensitivity (S) of conductive aerogels, they were subjected to a continuous compression process using the Instron compression tester, reaching up to 30% strain at a rate of 0.02 mm s$^{-1}$. Throughout this compression testing, the relative resistance changes of conductive aerogels were consistently monitored. Additionally, their stress–strain curves and $\sigma_{30}$ values were characterized. The S of the aerogel was subsequently calculated using Eq. (4).

### Joule heating of strain-insensitive conductive aerogels

To evaluate the Joule heating performance, the strain-insensitive conductive aerogels were first positioned on a two-electrode system. This system was then linked to either an electrochemical workstation (Autolab PGSTAT302N, Metrohm) or a portable battery setup. To measure the surface temperature, thermocouple probes (K Type Thermometer, Gain Express AZ Instruments) were attached to the sides of the aerogels. The electrochemical workstation enabled the application of varying voltages, ranging from 0.0 to 5.0 V. During cycling tests, the conductive aerogel was subjected to constant voltages of 1.0 and 1.5 V for 100-s intervals, followed by a 100-s pause.

### Thermal conductivity measurements of strain-insensitive conductive aerogels

Thermal conductivity measurements were carried out at room temperature using a heat flow meter (HFM-25, Thermtest Instruments). The temperature differentials were set at 17.5 °C and 32.0 °C. For these tests, the strain-insensitive conductive aerogel with the dimensions of 5.0 × 5.0 × 2.0 cm$^3$ was utilized.

### Characterization

The microstructures of conductive aerogels were investigated using a field emission scanning electron microscope (FESEM, Hitachi SU-70) operating at 1.0–2.0 kV for low, medium, and high-resolution imaging, equipped with an energy dispersive spectroscopy (EDS) for elemental analyses. All of the SEM samples were sputtered with a layer of AuPd (-1.0 nm) prior imaging. XPS were obtained by an X-ray photoelectron spectrometer (Kratos AXXIS UltraDLD) using a microfocused Al X-ray beam (100 μm, 25 W) with a photoelectron take-off angle of 90°.

### Construction of finite element (FE) models

The FE models of conductive aerogels were created using the commercial package ABAQUS/CAB 2020. The microstructures of high-, medium-, and low-density aerogels (at the same MXene/CNF/gelatin/ GA ratio of 80/20/0/–) were represented as two-dimensional periodic cellular structures with rectangular unit cells. The pore dimensions were approximated from the SEM images of experimental samples (Fig. 4c). The FE model structure was represented using plane strain elements within a 4 × 4 supercell, onto which periodic boundary conditions were imposed by relating the

displacements of nodes on opposing sides of the structure. To simulate the effects of the random imperfections observed in the SEM images of experimental samples, a constant number of wall discontinuities was seeded for all three FEA models, with the exact number chosen as a tuning parameter. By imposing 30% vertical compression on the supercells, the $\sigma_{30}$ values were extracted from 10 successive runs with differently seeded discontinuities. The localized stress profiles across the aerogel structures were extracted at both relaxed and compressed states.

## Reporting summary

Further information on research design is available in the Nature Portfolio Reporting Summary linked to this article.

## Data availability

The data used for model training is available from Zenodo repository: "Shrestha, S. *Data for: Machine Intelligence Accelerated Design of Conductive MXene Aerogels with Programmable Properties.* https://doi.org/10.5281/zenodo.10779454 (2024)" The data that support the plots within this paper and other finding of this study are available from the corresponding authors upon request. Source data are provided with this paper.

## Code availability

The Python code to implement the machine learning tasks within this study are available from GitHub (https://github.com/oshin71/Programmable-MXene-Aerogel).

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

## Acknowledgements

The authors acknowledge the financial support provided by the Start-Up Fund of University of Maryland, College Park (KFS No.: 2957431 to P.-Y.C.). Fundings for this research were provided by MOST-AFOSR Taiwan Topological and Nanostructured Materials Grant under Grant No. FA2386-21-1-4065 (KFS No.: 5284212 to P.-Y.C.), Energy Innovation Seed Grant from Maryland Energy Innovation Institute (MEI^2) (KFS No.: 2957597 to P.-Y.C.), and UMD Grand Challenges Team Project Grant (KFS No.: 2957821 to to P.-Y.C.). BioRender was used to create Fig. 1a, Fig. 2a, b, Fig. 3c, Fig. 5a, Supplementary Fig. 1, Supplementary Fig. 2.

## Author contributions

P.-Y.C. and S.S. conceived the project ideas and designed the experiments. T.C., S.S., Y.L., and J.M.L. carried out the synthesis and characterization of MXene nanosheets and CNF. T.C. and H.Y. designed the ML framework and implemented the ML tasks in Python. S.S. and M.M.K. collected the experimental data. K.J.B. and E.T. performed FE simulations and constructed FE models of conductive aerogels. P.-Y.C. and S.S. interpreted the results and co-wrote the manuscript. P.-Y.C., E.T., S.S., K.J.B., Y.L., T.C., J.M.L., H.Y., M.M.K., Z.T., and Y.L. were involved in the discussion. P.-Y.C., E.T., and Y.L. supervised this project.

## Competing interests

The authors declare no competing interests.
