## [Peer Review File · Nature Communications]

Machine Intelligence Accelerated Design of Conductive MXene Aerogels with Programmable PropertiesREVIEWER COMMENTS

Reviewer #1 (Remarks to the Author):

This manuscript incorporated the robotics-accelerated experimentation, machine learning, and simulation to expedite the tailored design of conductive aerogels. An automated pipetting robot is developed to fabricate 264 mixtures using four building blocks at different ratios/loadings, and 162 kinds of conductive aerogels are fabricated/characterized, enabling the construction of an artificial neural network prediction model. Then, conductive aerogels with high electrical conductivity, customized compression resilience, and pressure insensitivity is predicted, fabricated and used for wearable thermal management. This work is novel in the instruments and methods, but less important from the scientific point of view. I would like to recommend this manuscript for publication after it is revised taking into considerations the following points:

1. Aerogels are materials with ultra-low density. What is the effect of density on the electrical properties? How do the densities of aerogels vary with grade A, B and C?
2. Please list the descriptors used in the machine learning and explain the rationale.
3. The fabrication parameters (as listed in Figure S5) include only the ratios/loading, which are related to the composition. Other fabrication parameters, such as temperatures, pressures, should be included.
4. Page 6, paragraph 2. "Over 5300 data points are required, given a step size of 2.0wt% and four mixture loadings". Please illustrate how the number of 5300 is calculated based on the step size of 2.0wt%.
5. Page 9, in Fig. 1(b) and (c), which axis represents the amount of gelatin?
6. Page 12, "Finally, the database contained approximately 160,000 data points". Are the 160,000 data points employed in the prediction?

Reviewer #2 (Remarks to the Author):

The manuscript adeptly demonstrates the integration of automated sample testing via a robotic system with machine learning (ML) techniques to establish correlations between the compositions and properties of conductive aerogels. The topic's quality and novelty align well with the scope and expectations of Nature Communications. The concept is clearly and concisely introduced, the figures and videos are of high quality and aesthetically pleasing, the approach is scientifically robust, and the topic is of significant current interest to a broad scientific community, extending well beyond the scope of more specialized journals.

While I recommend publication without further revisions, I offer the following minor critiques to enhance the quality of this manuscript further:

Contextualization with recent work: Significant work has been published recently on automating experimentation in various contexts of materials development. The manuscript would benefit from summarizing these efforts, and comparing them with the proposed methods in terms of novelty, advantages, and potential disadvantages.

Exploration of known correlations: In the introduction, the authors mention that correlations between the ingredients are "largely" unexplored, but do not elaborate on those that are explored. Providing more details on these known correlations and potentially incorporating them into their experimentation could increase efficiency.

Comparison with traditional (experimental) approaches: The implementation of SHAP and FEA to elucidate the ML algorithm's reasoning is well-founded. However, it would be beneficial if the authors provided more context on the advantages of this approach compared to traditional experimental methods, such as Design of Experiments (DoE), optimization methods, and

experimentally validated finite element analysis (FEA) models, which might yield comparable results in potentially less time.

Quantitative comparison with DoE: The authors briefly mention DoE in the conclusion, claiming greater efficiency for their approach. This assertion would be more compelling if supported by quantitative data, such as the number of experimental runs required by a DoE design to achieve a similar level of information and predictive capability.

Writing improvements: Address minor grammatical issues, such as inconsistencies in parenthesis usage (e.g., "silk nanofibrils, chitosan"). Also, avoid repetitive language, like the repetition of 'conductivity' in the very first sentence. Some statements are vague, such as "Multiple blind tests were performed by different researchers to maintain unbiased evaluations." Specifying the number of tests would provide clarity.

Methodological details: Clarify how the freeze-drying process is automated, if at all. While there is some context in the Supporting Information, a brief mention in the main document would be beneficial.

Clarification on testing procedures: The results suggest that conductivity was tested after compressive stress testing. It would be helpful if the authors clarified whether new samples were produced for each test or if the same samples were reused (Page 11).

Reviewer #3 (Remarks to the Author):

This manuscript reported the conductive hydrogels inks with different components by machine learning and freeze drying. The resultant aerogels with the desired conductivity and compressibility are obtained by designing different ratios. However, it is common to prepare both conductive and elastic aerogels by introducing elastic CNF and conductive MXene, many studies have explored MXene/CNF systems including their conductivity and compressibility. In this experiment, machine learning only replaced the feeding and matching process, but could not independently analyze the results to improve the formula. It may be far from people's expectations to the significance of machine learning in related areas. Compared to previous reports, there is no enough novelty to ensure its publication in nature communications.

1. MXene/CNF aerogels have been widely reported, and optimization schemes can be selected based on the establishment of conductive thresholds and elastic systems.
2. Machine learning replaces the human formulation of different proportions of the mixture, the lack of a reasonable screening scheme - analysis - self-correction - the process of drawing conclusions.
3. Why choose MXene/CNF system, how to solve the oxidation problem and the aggregation problem introduced by CNF of MXene?
4. There are some typos and grammar mistakes, the authors should be more careful.

Responses to Reviewers' Comments and Revisions Made

(Blue type: Reviewers' remarks; Black type: Our response)

Reviewer #1

This manuscript incorporated the robotics-accelerated experimentation, machine learning, and simulation to expedite the tailored design of conductive aerogels. An automated pipetting robot is developed to fabricate 264 mixtures using four building blocks at different ratios/loadings, and 162 kinds of conductive aerogels are fabricated/characterized, enabling the construction of an artificial neural network prediction model. Then, conductive aerogels with high electrical conductivity, customized compression resilience, and pressure insensitivity is predicted, fabricated and used for wearable thermal management.

Response R1.1

We would like to express our gratitude to Reviewer #1 for evaluating our work and providing valuable comments. We have considered all their comments and revised the manuscript and Supporting Information accordingly.

This work is novel in the instruments and methods, but less important from the scientific point of view.

Response R1.2

We appreciate that Reviewer #1 recognized the novelty of instruments and methods. Besides this novelty, we would like to provide three additional perspectives to further highlight our work.

In the revised manuscript, we have compared our robotics/ML-integrated workflow with the state-of-the-art works, as summarized in **Table S11**, **S12**, and **S13**.

1. Construction of a prediction model with high accuracy across the entire parameter space.

Our work successfully demonstrated a versatile workflow that combined collaborative robotics and AI/ML predictions, enabling efficient development and optimization of various functional nanocomposites (e.g., conductive aerogels) with user-designated properties. By incorporating collaborative robotic technologies, we can significantly enhance the speed of acquiring high-quality data points and minimize human variations. Meanwhile, the collection of data points for the next rounds of experiments can be guided by the AI/ML predictions.

As detailed in **Table S11**, the state-of-the-art works typically focus on pinpointing an optimal set of fabrication parameters to maximize device performance or identifying precise conditions necessary for successful chemical reactions. While some of these works also utilize high-throughput robotic platforms to navigate a parameter space with multiple degrees of freedom (DOFs), they do not intend to examine data points that stray from the primary objectives, such as determining the most effective parameter combination or sequence of reaction steps. Consequently, if the design requirements shift – for instance, the synthesis of a new chemical compound – new round(s) of optimization experiments would be necessary. This requirement for additional experimentation may not align well with demands for rapid customization. In contrast to traditional methods, our integrated workflow that combines collaborative robotics with SVM classification, active learning loops, and data augmentation, resulted in a prediction model that

consistently delivers high accuracy across the entire parameter space. Consequently, to respond to diverse design requests for customization, our prediction model can automatically identify the optimal fabrication parameters, eliminating the need for repeated optimization experiments.

2. Automatic design of conductive aerogels with programmable properties. Through the robotics/ML-integrated workflow, the prediction model can perform two-way design tasks: (1) predicting the physicochemical properties of conductive aerogels based on their fabrication parameters and (2) automating the inverse design of conductive aerogels to align with specific property requirements. The prediction model is well-suited for crafting conductive aerogels with customizable mechanical and electrical properties. Given the laborious nature of self-assembly and the absence of robotic systems available for fully automating freeze drying processes, our prediction model can directly identify viable and optimal fabrication parameters, without the need for iterative optimization experiments. Compared with the state-of-the-art works of conductive aerogels (see **Table S12**), our automatic design method demonstrates superior efficiency in discovering conductive aerogels with programmable properties.

3. Data-driven elucidation of design principles in complex material systems. **Table S13** presents a comparison of our work with multiple studies focusing on MXene-based aerogels, including MXene/CNT, MXene/rGO, MXene/ANF, and MXene/CNF. Traditionally, these state-of-the-art works explore the intricate correlations between fabrication parameters and aerogel properties through the one-factor-at-a-time (OFAT) design of experiment (DoE) method. The OFAT DoE method has yielded diverse design insights across different MXene-based aerogel systems. For example, Zeng *et al.*, Xu *et al.*, Jiang *et al.*, and Wang *et al.* groups discovered that increasing the MXene loading decreased the compressive strength and electrical resistance of conductive aerogels. In contrast, our research harnesses the synergy of collaborative robotics and advanced AI/ML predictive analytics. By constructing a robust prediction model followed by model interpretation (e.g., SHAP), we can investigate several component–property correlations and generalize design principles with statistical significance, offering a more systematic understanding of complex material systems.

Revision Made

Page 28 of the revised manuscript

Compared with the state-of-the-art works of conductive aerogels, our robotics/ML-integrated workflow demonstrates multiple advancements, as summarized in **Table S11**,⁷³⁻⁷⁵ **S12**,^{2,8,19,20,29,33,76-85} and **S13**.^{8,19,20,29,33,76-82} The first advancement is to construct a prediction model with high accuracy across the entire parameter space. As detailed in **Table S11**, the state-of-the-art works typically focus on pinpointing an optimal set of fabrication parameters to maximize device performance or identifying precise conditions necessary for successful chemical reactions. While some of these works also utilize high-throughput robotic platforms to navigate a parameter space with multiple degrees of freedom (DOFs), they do not intend to examine data points that stray from the primary objectives, such as determining the most effective parameter combination or sequence of reaction steps. Consequently, if the design requirements shift – for instance, the synthesis of a new chemical compound – new round(s) of optimization experiments would be necessary. This requirement for additional experimentation may not align well with demands for rapid customization. In contrast to traditional methods, our integrated workflow that combines collaborative robotics with SVM classification, active learning loops, and data augmentation,

resulted in a prediction model that consistently delivers high accuracy across the entire parameter space. Consequently, to respond to diverse design requests for customization, our prediction model can automatically identify the optimal fabrication parameters, eliminating the need for repeated optimization experiments.

The second advancement is to automate the design of conductive aerogels with programmable properties. Through the robotics/ML-integrated workflow, the prediction model can perform two-way design tasks: (1) predicting the physicochemical properties of conductive aerogels based on their fabrication parameters and (2) automating the inverse design of conductive aerogels to align with specific property requirements. The prediction model is well-suited for crafting conductive aerogels with customizable mechanical and electrical properties. Given the laborious nature of self-assembly and the absence of robotic systems available for fully automating freeze drying processes, our prediction model can directly identify viable and optimal fabrication parameters, without the need for iterative optimization experiments. Compared with the state-of-the-art works of conductive aerogels (see **Table S12**), our automatic design method demonstrates superior efficiency in discovering conductive aerogels with programmable properties.

The third advancement is to elucidate data-driven design principles in complex material systems. **Table S13** presents a comparison of our work with multiple studies focusing on MXene-based aerogels, including MXene/CNT, MXene/rGO, MXene/ANF, and MXene/CNF. Traditionally, these state-of-the-art works explore the intricate correlations between fabrication parameters and aerogel properties through the one-factor-at-a-time (OFAT) design of experiment (DoE) method. The OFAT DoE method has yielded diverse design insights across different MXene-based aerogel systems. For example, Zeng *et al.*, Xu *et al.*, Jiang *et al.*, and Wang *et al.* groups discovered that increasing the MXene loading decreased the compressive strength and electrical resistance of conductive aerogels.^{33,76,80,81} In contrast, our research harnesses the synergy of collaborative robotics and advanced AI/ML predictive analytics. By constructing a robust prediction model followed by model interpretation (e.g., SHAP), we can investigate several component–property correlations and generalize design principles with statistical significance, offering a more systematic understanding of complex material systems.

Table S11. Comparison of our AI/ML framework with the state-of-the-art works.

Independent Variables (# of DOFs)	Optimization/Prediction Targets (# of Property Labels)	Multi-Property Optimization	Sampling Method	Experiments Required for New Design Requests	Sample Preparation and Characterization Platforms	Data Collection Rate	Machine Learning Algorithms	Ref.
Fuel-to-oxidizer ratio, fuel blend, total concentration, anneal temperature (4 DOFs)	Determining the optimal fabrication parameters for high-conductivity palladium films at low annealing temperatures (2 property labels)	Yes	Bayesian optimization (w/ qEHVI acquisition function)	Possible	Ada: Precision 4-axis laboratory robot (N9)/6-axis collaborative robot (UR5e)	2 data per hour	Gaussian process regression	1
Column count, column outer radius, column thickness, twisted angle (4 DOFs)	Pinpointing the optimal structural parameters for high compression toughness (1 property label)	No	Bayesian optimization	Possible	BEAR: 3D printers/robotic arm/scale/universal testing machine	64 data per day	Gaussian process regression	2
Reagent injection sequence, reaction time length, volume (>40 DOFs)	Determining the optimal reaction parameters for synthesizing hetero-nanostructures with high λ_{AP} , R_{PV} , I_{PL} (3 property labels)	Yes	Reinforcement learning	Possible	AlphaFlow: A self-driven fluidic lab consisting of reagent injection, droplet oscillation, optical sampling, phase separation, and waste collection.	2 data per hour	Reinforcement learning	3
MXene loading, CNF loading, gelatin loading, GA loading, mixture loading (3 DOFs)	Aiming to predict across the full parameter space to produce conductive aerogels with customized mechanical and electrical properties (2 property labels)	Yes	Active learning (w/ A score acquisition function)	No	OT-2 robot and UR5e-automated compression tester	20 data per 2.5 days	Support-vector machine / artificial neural network / data augmentation	This work

Table S12. Comparison of our robotics/ML-integrated workflow with the state-of-the-art works regarding the production of conductive MXene aerogels.

Design Strategy	Aerogel Composition	Aerogel Density (mg cm^{-3})	σ_{30} (kPa)	Electrical Property ($-$)	Electrical Property Measurement	Ref.
Quaternary Aerogel Systems with MXene/CNF						
Robotics/ML-integrated workflow	MXene/CNF/gelatin/GA	2.0 – 14.0	0.05 – 16.1	$1.4 - 10^{10} \Omega$	Two-electrode testing system	This work
Binary Aerogel Systems with MXene/CNF						
Design of experiment	MXene/CNF	4.0	0.5	$2.8 \times 10^1 \text{ S cm}^{-1}$	Four-point probe	4
		4.0	0.8	$4.0 \times 10^{-1} \text{ S cm}^{-1}$		
Design of experiment	MXene/CNF	50	7.0	$2.6 \times 10^1 \Omega \text{ cm}$	Two-electrode testing system	5
		50	3.4	$1.4 \times 10^1 \Omega \text{ cm}$		
Ternary Aerogel Systems with MXene/CNF						
Design of experiment	MXene/CNF/CNT	7.5	0.4	$2.4 \times 10^3 \text{ S cm}^{-1}$	Four-point probe	6
Design of experiment	MXene/CNF/PMDI	18.0	40.0	$1.1 \times 10^0 \Omega \text{ cm}$	Four-point probe	7
Other MXene Aerogel Systems						
Design of experiment	MTMS treated bacterial cellulose/MXene	6.2	1.6	$1.2 \times 10^2 \text{ S cm}^{-1}$	Four-point probe	8
Design of experiment	MXene/CNT	9.1	0.5	$4.5 \times 10^2 \text{ S cm}^{-1}$	Four-point probe	9

Design Strategy	Aerogel Composition	Aerogel Density (mg cm ⁻³)	σ_{30} (kPa)	Electrical Property (-)	Electrical Property Measurement	Ref.
Design of experiment	PGPDMS/MXene	9.9	0.1	–	–	10
Design of experiment	MXene/rGO	12.2	7.5	3.6×10^1 S cm ⁻¹	Three-electrode setup	11
Design of experiment	MXene	12.6	1.0	5.4×10^1 S cm ⁻¹	Two-electrode testing system	12
Design of experiment	MXene/ANF	25.0	17.0	1.0×10^4 Ω cm	Two-electrode testing system	13
Design of experiment	MXene/CNT/ANF	42.0	2.3	2.7×10^0 Ω cm	Four-point probe	14
		42.0	0.8	2.0×10^{-1} Ω cm	Four-point probe	
Design of experiment	MXene/Waterborne polyurethane (WPU)	–	3.0	–	–	15
Other Conductive Aerogel Systems						
Design of experiment	rGO	2.3	6.0	1.6×10^1 S cm ⁻¹	Four-point probe	16
Design of experiment	rGO	8.5	8.2	4.3×10^1 S cm ⁻¹	Four-electrode testing system	17
Design of experiment	GO/PVA	10.0	5.0	3.0×10^0 S cm ⁻¹	Source meter	18
Design of experiment	GO	25.0	1.0	9.7×10^0 S cm ⁻¹	Source meter	19

Abbreviations

rGO: Reduced graphene oxide

GO: Graphene oxide

MTMS: Methyltrimethoxysilane

CNT: Carbon nanotube

PGPDMS: Glycidoxypropyldimethoxymethylsilane

PVA: Poly(vinyl alcohol)

PMDI: Poly((phenyl isocyanate)-co-formaldehyde)

ANF: Aramid nanofiber

WPU: Waterborne polyurethane

Table S13. Comparison of our AI/ML and data-driven approach with the state-of-the-art works regarding design insight elucidation.

Insight Elucidation Strategy	Building Block(s)	Discovery Method(s)	Design Insight(s)	Ref.
Quaternary Aerogel Systems with MXene/CNF (Our Work)				
AI/ML and data-driven approach	MXene/CNF/gelatin/GA	Spearman's analysis, SHapley Additive exPlanations (SHAP) model interpretation	 1. The mixture loading had the most significant impact on the mechanical properties of conductive aerogels. 2. The MXene loading had the most significant impact on the electrical properties of conductive aerogels. 	This work
Binary Aerogel Systems with MXene/CNF				
Design of experiment	MXene/CNF	Experimental observation	 1. The addition of 33 wt.% CNF led to a 350% increase in the compressive moduli of conductive aerogels. 2. The addition of 17 wt.% CNF had a minimal impact on the conductivity of conductive aerogels, while the addition of 50 wt.% CNF significantly decreased the conductivity of these aerogels. 	4
Design of experiment	MXene/CNF	Experimental observation	 1. Increasing the MXene loading decreased the compressive strength of conductive aerogels under 33% strain. 2. Increasing the MXene loading from 0.7 to 14 wt.% decreased the resistance of conductive aerogels from 26.3 to 13.8 Ω. 	5
Ternary Aerogel Systems with MXene/CNF				
Design of experiment	MXene/CNF/CNT	Experimental observation	 1. The presence of CNF plays a crucial role in maintaining the structural integrity of aerogels. 	6
Design of experiment	MXene/CNF/PMDI	Experimental observation	 1. Adding 30 wt.% CNFs increased the compressive modulus of MXene/CNF aerogels by 155%. 2. The mechanical strength of MXene/CNF aerogels was improved by chemically crosslinking them with PMDI. 	7
Other MXene Aerogel Systems				

Insight Elucidation Strategy	Building Block(s)	Discovery Method(s)	Design Insight(s)	Ref.
Design of experiment	MTMS treated bacterial cellulose/MXene	Experimental observation	 1. Aerogels fabricated from a 1:1 ratio of silylated bacterial cellulose to MXene maintained a residual height of >95% and exhibited a compressive strength of 6 kPa under 50% strain. 2. Conversely, aerogels produced from a 1:1 ratio of untreated bacterial cellulose and MXene retained a residual height of approximately 70% and displayed a compressive strength of 5 kPa. 	8
Design of experiment	MXene/CNT	Experimental observation	 1. An aerogel, produced from a 95:5 ratio of MXene to CNT, exhibited a plastic deformation of 4.2% and an electrical conductivity of 450 S m⁻¹. 2. Another aerogel, produced from a 60:40 ratio of MXene to CNT, showed a plastic deformation of 2.1% and an electrical conductivity of 100 S m⁻¹. 	9
Design of experiment	PGPDMS/MXene	Experimental observation	 1. The integration of PGPDMS into the MXene interlayers produced an ultra-soft aerogel. 2. At aerogel densities of 7.0 and 10 mg cm⁻³, the compressive moduli of the PGPDMS/MXene aerogels were 58 and 140 Pa respectively. 	10
Design of experiment	MXene/rGO	Experimental observation	 1. As the MXene loading increased from 25 to 75 wt.%, the aerogels' maximum strain decreased from 95% to 60%. 2. When the annealing temperature rose from room temperature to 300 °C, both the maximum strain and the conductivity of conductive aerogels increased. 	11
Design of experiment	MXene	Experimental observation	 1. The compressive stress and electrical conductivity of MXene aerogels both increased with the rise in aerogel density. 2. Factors such as sheet alignment, pore size, and overall aerogel microstructure influenced the aerogels' electrochemical properties. 	12
Design of experiment	MXene/ANF	Experimental observation	 1. As the MXene loading increased from 30, 50, to 70 wt.%, the compressive stress of MXene/ANF aerogels decreased by 8.6%, 25.3%, and 30.1%, respectively. 2. The aerogel's resistance also displayed a similar trend in relation to the MXene loading. 	13
Design of experiment	MXene/CNT/ANF	Experimental observation	 1. As the CNT loading increased, the mechanical properties of the composite aerogels decreased. 2. On the other hand, a higher CNT loading improved the electrical conductivity of these composite aerogels. 	14

Insight Elucidation Strategy	Building Block(s)	Discovery Method(s)	Design Insight(s)	Ref.
Design of experiment	MXene/Waterborne polyurethane (WPU)	Experimental observation	 1. The composite aerogel, made from functionalized cellulose nanocrystal (f-NCC), MXene, and polyurethane, withstood 100 compression–relaxation cycles, maintaining 76.2% of the initial maximum stress. 2. The wood-like microstructure, interconnected network, and interactions between the f-NCC, MXene, and PU matrix improved the compressibility and elasticity of the composite aerogels. 	16

I would like to recommend this manuscript for publication after it is revised taking into considerations the following points:

Response R1.3

We would like to thank Reviewer #1's support and recommendation.

Comment 1: Aerogels are materials with ultra-low density. What is the effect of density on the electrical properties? How do the densities of aerogels vary with grade A, B and C?

Response R1.4

As illustrated in **Fig. S17**, the SHAP model interpretation method was utilized to assess the influence of MXene loading, CNF loading, gelatin loading, GA loading, and mixture loading on the initial resistances of conductive aerogels (R_0). As detailed in **Fig. S18**, the mixture loading notably correlated positively with the aerogel density. The SHAP values for MXene loading on R_0 ranged from -0.94 to $+0.97$, demonstrating the most substantial variability and pronounced impact compared to other factors: CNF loading (-0.41 to $+0.65$), gelatin loading (-0.83 to $+0.94$), GA loading (-0.24 to $+0.36$), and mixture loading (-0.35 to $+1.00$). These SHAP results highlight that increasing the MXene loading significantly lowered the R_0 of conductive aerogels, marking it as the most influential factor. Similarly, increasing the mixture loading, which corresponds to increased aerogel density, had a similar effect on lowering the R_0 of conductive aerogels. However, the influence of mixture loading (or aerogel density) was statistically less significant than that of MXene loading, and it showed a high degree of correlation with the MXene/CNF/gelatin/GA ratio.

To validate the SHAP-identified correlations, additional experiments were conducted in **Fig. S19a**. With the MXene/CNF/gelatin/GA ratio fixed at 11/77/12/+, conductive aerogels consistently exhibited high R_0 values with minimal variations across different mixture loadings (ranging from 10.0 to 2.5 mg mL⁻¹). The corresponding SEM images are provided in **Fig. S19b**. Conversely, when the MXene/CNF/gelatin/GA ratio was set to 61/28/11/–, conductive aerogels displayed a significant decrease in their R_0 values but followed a similar trend across different mixture loadings with values ranging from 2.1 to 17.3 Ω . The corresponding SEM images are provided in **Fig. S19c**. In contrast, the conductive aerogels prepared from the MXene/CNF/gelatin/GA ratio of 43/42/15/–, showed pronounced variations in R_0 values spanning from 7.5 to 228.4 Ω at different mixture loadings. The corresponding SEM images are provided in **Fig. S19d**. These results show that the MXene loading exhibited a higher impact on the R_0 of conductive aerogels than the mixture loading.

Revision Made

Page 22 of the revised manuscript

On the other hand, **Fig. S17** shows that the SHAP values for MXene loading on R_0 ranged from -0.94 to $+0.97$, demonstrating the most substantial variability and pronounced impact compared to other factors: CNF loading (-0.41 to $+0.65$), gelatin loading (-0.83 to $+0.94$), GA loading (-0.24 to $+0.36$), and mixture loading (-0.35 to $+1.00$). These SHAP results highlight that increasing the MXene loading significantly lowered the R_0 of conductive aerogels, marking it as the most influential factor. As detailed in **Fig. S18**, the mixture loading notably correlated

positively with the aerogel density. Similarly, increasing the mixture loading, which corresponds to increased aerogel density, had a similar effect on lowering the R_0 of conductive aerogels. However, the influence of mixture loading (or aerogel density) was statistically less significant than that of MXene loading, and it showed a high degree of correlation with the MXene/CNF/gelatin/GA ratio.

To validate the SHAP-identified correlations, additional experiments were conducted in **Fig. S19a**. With the MXene/CNF/gelatin/GA ratio fixed at 11/77/12/+, conductive aerogels consistently exhibited high R_0 values with minimal variations across different mixture loadings (ranging from 10.0 to 2.5 mg mL⁻¹). The corresponding SEM images are provided in **Fig. S19b**. Conversely, when the MXene/CNF/gelatin/GA ratio was set to 61/28/11/–, conductive aerogels displayed a significant decrease in their R_0 values but followed a similar trend across different mixture loadings with values ranging from 2.1 to 17.3 Ω . The corresponding SEM images are provided in **Fig. S19c**. In contrast, the conductive aerogels prepared from the MXene/CNF/gelatin/GA ratio of 43/42/15/–, showed pronounced variations in R_0 values spanning from 7.5 to 228.4 Ω at different mixture loadings. The corresponding SEM images are provided in **Fig. S19d**. These results show that the MXene loading exhibited a higher impact on the R_0 of conductive aerogels than the mixture loading.

Page 20 of Supporting Information

Fig. S17. SHAP values of MXene loading, CNF loading, gelatin loading, and GA loading, and mixture loading on the R_0 values of conductive aerogels.

Fig. S18. Positive relationships between aerogel density and mixture loading. We characterized the densities of conductive aerogels at four different MXene/CNF/gelatin/GA ratios and four mixture loadings, which ranged from 2.5 to 10.0 mg mL⁻¹. Data are presented as mean \pm s.d., $n = 3$, with each independent experiment marked by an open black circle.

Fig. S19. Dependence of electrical resistances and aerogel microstructures on mixture loading. (a) Correlations between aerogel electrical resistance and mixture loading. (b) SEM images of conductive aerogels at the MXene/CNF/gelatin/GA ratio of 11/77/12/+ at different mixture loadings. (c) SEM images of conductive aerogels at the MXene/CNF/gelatin/GA ratio of 61/28/11/- at different mixture loadings. (d) SEM images of conductive aerogels at the MXene/CNF/gelatin/GA ratio of 43/42/15/- at different mixture loadings.

Comment 2: Please list the descriptors used in the machine learning and explain the rationale.

Response R1.5

We thank Reviewer #1's insightful comment. In the revised manuscript, we have explained (1) the rationale of each unit used in the ML framework (see **Note S3**) and (2) summarized the descriptors (i.e., labels) used in the prediction model (see **Table S1**).

Revision Made

Page 7 of the revised manuscript

The rationale of each phase is detailed in **Note S3**. **Table S1** summarized the descriptors (i.e., labels) used in the prediction model. **Fig. S7** compares the prediction accuracy of models using different labels to represent the mechanical properties of conductive aerogels.

Page 30 of Supporting Information

Note S3. Multi-stage AI/ML framework.

To construct a high-accuracy prediction model, an AI/ML framework was developed and had three critical phases: (1) establishing a feasible parameter space, (2) implementing active learning loops, and (3) synthesizing virtual data points. We provide in-depth justifications for the algorithmic choices in the AI/ML framework and elucidate the reasons why the selected algorithms outperform traditional or simplistic methods in terms of efficiency, accuracy, and scalability.

SVM classifier: Within the AI/ML framework, the SVM classifier acted as a crucial filtering unit for the prediction model, ensuring the model-suggested MXene/CNF/gelatin/GA ratios that produced *A*-grade conductive aerogels. In contrast to other systems rich in data, the fabrication of conductive aerogels is time-consuming and labor-intensive. In the absence of the SVM classifier, the prediction model could suggest the MXene/CNF/gelatin/GA ratios that result in lower-grade conductive aerogels, which would not be efficient for collecting high-quality data points. Therefore, a high-accuracy SVM classifier is critical to ensure the consistent production of *A*-grade conductive aerogels throughout the active learning loops.

ANN committee: As indicated in **Fig. 2e**, the prediction models based on alternative algorithms struggled to accurately predict the σ_{30} and R_0 values of conductive aerogels based on their MXene/CNF/gelatin/GA ratios and mixture loadings. In contrast, our prediction model, which incorporated an ANN committee, demonstrated the lowest MAE of 1.5 kPa (for σ_{30}) and MRE of 18.4% (for R_0) after the active learning loops. These results clearly show that the ANN committee is required to accurately predict the mechanical and electrical properties of conductive aerogels in a complex, non-linear system with multiple DOFs.

Data augmentation: To overcome the overfitting challenges upon the use of a small dataset, we employed a data augmentation method, specifically the UIP method, to synthesize virtual data points in the vicinity of real data points collected during the active learning loops. By feeding both real and virtual data points into the ANN committee, the prediction model demonstrated higher predictive accuracy, as evidenced by lower MRE and MAE values, as demonstrated in **Fig. 2f**.

In conclusion, the multi-stage AI/ML framework, composed of an SVM classifier, active learning loops (incorporating an ANN committee), and data augmentation, can synergistically address the data scarcity challenges encountered in the field of conductive aerogels, where the collection of high-quality data points is labor-intensive and time-consuming.

Table S1. Description of independent and dependent variables.

Independent Variable	Unit	Physical Meaning Description
MXene Loading	wt.%	Weight percentage of MXene nanosheets in an aqueous mixture
CNF Loading	wt.%	Weight percentage of CNFs in an aqueous mixture
Gelatin Loading	wt.%	Weight percentage of gelatin in an aqueous mixture
GA Loading	+/-	Options of adding GA into an aqueous mixture (“+” with GA addition; “-” without GA addition)
Mixture Loading	mg mL ⁻¹	Solid content in an aqueous mixture
Dependent Variable	Unit	Physical Meaning Description
R_0	Ω	Initial electrical resistance under 0% strain
σ_{30}	kPa	Compressive stress under 30% strain*

*There are two reasons of selecting σ_{30} as one of the dependent variables. First, several low-density aerogels were too fragile and could not sustain the compressive strains over 30%. Second, as shown in **Fig. S7**, taking multiple points along the entire stress–strain curves as the training data points led to a prediction model with a higher MAE value of 4.5 kPa. On the other hand, when the σ_{30} value was used as the mechanical labels for training data points, the prediction model demonstrated a lower MAE value of 1.1 kPa.

Fig. S7. Comparison of prediction models using different mechanical labels. When the entire stress-strain curve was used as the mechanical labels for training data points, the prediction model demonstrated a higher MAE value of 4.5 kPa. On the other hand, when the value was used as the mechanical labels for training data points, the prediction model demonstrated a lower MAE value of 1.1 kPa.

Comment 3: The fabrication parameters (as listed in Figure S5) include only the ratios/loading, which are related to the composition. Other fabrication parameters, such as temperatures, pressures, should be included.

Response R1.6

We appreciate the comments from Reviewer #1, and we believe that Reviewer #1 might have intended to point to **Fig. S5** (Previously **Fig. S4**), not **Fig. S8** (Previously **Fig. S5**). Regarding the fabrication processes of conductive aerogels in this study, both freeze drying temperature and pressure were held constant at $-80\text{ }^{\circ}\text{C}$ and 0.3 Pa , respectively. Therefore, they were not considered as independent variables. To eliminate any further misunderstanding, we have revised the captions of **Fig. S5** and **Fig. S8** to indicate that freeze drying temperature and pressure were consistent across all experiments.

Revision Made

Page 7 of Supporting Information

Fig. S5. Tunable mechanical and electrical properties of conductive aerogels. The mechanical and electrical properties of conductive aerogels, such as σ_{30} and R_0 , showed complex and nonlinear correlations with the MXene/CNF/gelatin/GA ratios and the solid contents of aqueous mixtures (abbreviated as mixture loadings). **(a)** Stress–strain curves of eight conductive aerogels with different MXene/CNF/gelatin/GA ratios and mixture loadings. **(b)** R_0 values of eight conductive MXene aerogels with different MXene/CNF/gelatin/GA ratios and mixture loadings. It is worth mentioning that, with regard to the fabrication processes of conductive aerogels, both freeze drying temperature and pressure were held constant at $-80\text{ }^{\circ}\text{C}$ and 0.3 Pa , respectively. Therefore, they were not considered as independent variables.

Page 11 of Supporting Information

Fig. S8. Feasible parameter space of conductive aerogels. **(a)** Binary maps representing the feasible parameter space of obtaining *A*-grade conductive aerogels. **(b)** As the mixture loading decreased from 10.0 to 2.5 mg mL^{-1} , the area of the feasible parameter space decreased from 83.7% to 48.6% , respectively. It is worth mentioning that, with regard to the fabrication processes of conductive aerogels, both freeze drying temperature and pressure were held constant at $-80\text{ }^{\circ}\text{C}$ and 0.3 Pa , respectively. Therefore, they were not considered as independent variables.

Comment 4: Page 6, paragraph 2. “Over 5300 data points are required, given a step size of $2.0\text{wt}\%$ and four mixture loadings”. Please illustrate how the number of 5300 is calculated based on the step size of $2.0\text{wt}\%$.

Response R1.7

To compare our ML framework with the one-factor-at-a-time (OFAT) design of experiment (DoE) method, we calculated the number of data points needed to cover the entire design space using **Equations S1** and **S2**,

$$S_x = \frac{(n!)}{k!(x-1)!} \quad (\text{S1})$$

$$n = x - 1 + k \quad (\text{S2})$$

, where S_x denotes the number of data points, x is the number of DOFs, and k is the step size. The Python code is available on GitHub at <https://github.com/oshin71/Programmable-MXene-Aerogel/blob/main>. Using the step size of 2.0 wt.%, the OFAT DoE method requires 5,300 data points. As depicted in **Fig. S6**, the number of data points is inversely correlated with the step size.

Revision Made

Page 6 of the revised manuscript

To establish a comprehensive database linking the fabrication parameters with the end properties of conductive aerogels, over 5,300 data points are required, given a step size of 2.0 wt.% and four mixture loadings (see our estimation in **Note S2** and **Fig. S6**).

Page 29 of Supporting Information

Note S2. Estimated number of experiments required to build an extensive dataset for conductive aerogels.

To compare our ML framework with the one-factor-at-a-time (OFAT) design of experiment (DoE) method, we calculated the number of data points needed to cover the entire design space using **Equations S1** and **S2**,

$$S_x = \frac{(n!)}{k! \cdot (x-1)!} \quad (\text{S1})$$

$$n = x - 1 + k \quad (\text{S2})$$

, where S_x denotes the number of data points, x is the number of DOFs, and $1/k$ is the step size. The Python code is available on GitHub at <https://github.com/oshin71/Programmable-MXene-Aerogel/blob/main>.

For the composition labels, three DOFs ($x = 3$) are identified for the fabrication of conductive aerogels. For the loading label of conductive aerogels, one DOF is identified. Considering a step size of 2.0 wt.% for the MXene, CNF, and gelatin loadings ($\frac{1}{k} = 0.02$), two options (+/-) for the GA incorporation, and four mixture loadings, a total number of experiments required to construct an extensive dataset is estimated to be 5,300. In addition, two property labels, including σ_{30} and R_0 , need to be collected for each conductive aerogel. Using the step size of 2.0 wt.%, the OFAT DoE method requires 5,300 data points. As depicted in **Fig. S6**, the number of data points is inversely correlated with the step size.

Fig. S6. Relationship between step size and number of data points required for an extensive dataset.

Comment 5: Page 9, in Fig. 1(b) and (c), which axis represents the amount of gelatin?

Response R1.8

Once the weight percentages of MXene nanosheets and CNFs are established, the weight percentage of gelatin is automatically determined. As illustrated in **Fig. S9**, while it is possible to plot the feasible parameter space using alternate component combinations (such as MXene–gelatin or CNF–gelatin), we think the plots with the axes of MXene and CNF loadings are clearer to see the feasible parameter space.

Revision Made

Page 8 of the revised manuscript

Fig. S9 illustrates similar data distribution plots using the MXene and gelatin loadings as the axes.

Fig. S9. Data distribution plots with MXene and gelatin loadings as the axes. (a) 264 MXene/CNF/gelatin aerogels with different grades based on their structural integrity and monolithic nature. (b) Four heatmaps showcasing the possibilities of producing *A*-grade conductive aerogels at specific MXene/CNF/gelatin ratios and mixture loadings. (c) Binary maps representing the feasible parameter space of obtaining *A*-grade conductive aerogels.

Comment 6: Page 12, “Finally, the database contained approximately 160,000 data points”. Are the 160,000 data points employed in the prediction?

Response R1.9

In this work, we collected 162 real data points during 8 active learning loops. Afterwards, to improve model training efficiency and counteract potential overfitting, the User Input Principle (UIP) method was adopted to synthesize virtual data points in a 1-to-1,000 ratio, thus leading to the generation of ~160,000 virtual data points. Afterward, both real and virtual data points were utilized as training data points for constructing the ANN model using 4-fold cross-validation.

To avoid further confusion, we have provided detailed description in the revised manuscript.

Revision Made

Page 13 of the revised manuscript

In this work, we collected 162 real data points during 8 active learning loops. Afterwards, to improve model training efficiency and counteract potential overfitting, the UIP method was adopted to synthesize virtual data points in a 1-to-1,000 ratio, thus leading to the generation of ~160,000 virtual data points. Afterward, both real and virtual data points were utilized as training data points for constructing the ANN model using 4-fold cross-validation.

Reviewer #2

The manuscript adeptly demonstrates the integration of automated sample testing via a robotic system with machine learning (ML) techniques to establish correlations between the compositions and properties of conductive aerogels. The topic's quality and novelty align well with the scope and expectations of Nature Communications. The concept is clearly and concisely introduced, the figures and videos are of high quality and aesthetically pleasing, the approach is scientifically robust, and the topic is of significant current interest to a broad scientific community, extending well beyond the scope of more specialized journals.

While I recommend publication without further revisions, I offer the following minor critiques to enhance the quality of this manuscript further:

Response R2.1

We would like to express sincere gratitude to Reviewer #2 for thoughtful review and positive feedback on our work. We have carefully considered each comment and have made the necessary revisions to the manuscript accordingly.

Comment 1: Contextualization with recent work: Significant work has been published recently on automating experimentation in various contexts of materials development. The manuscript would benefit from summarizing these efforts, and comparing them with the proposed methods in terms of novelty, advantages, and potential disadvantages.

Response R2.2

In the revised manuscript, we have compared our robotics/ML-integrated workflow with the state-of-the-art works, as summarized in **Table S11**, **S12**, and **S13**.

1. Construction of a prediction model with high accuracy across the entire parameter space. Our work successfully demonstrated a versatile workflow that combined collaborative robotics and AI/ML predictions, enabling efficient development and optimization of various functional nanocomposites (e.g., conductive aerogels) with user-designated properties. By incorporating collaborative robotic technologies, we can significantly enhance the speed of acquiring high-quality data points and minimize human variations. Meanwhile, the collection of data points for the next rounds of experiments can be guided by the AI/ML predictions.

As detailed in **Table S11**, the state-of-the-art works typically focus on pinpointing an optimal set of fabrication parameters to maximize device performance or identifying precise conditions necessary for successful chemical reactions. While some of these works also utilize high-throughput robotic platforms to navigate a parameter space with multiple degrees of freedom (DOFs), they do not intend to examine data points that stray from the primary objectives, such as determining the most effective parameter combination or sequence of reaction steps. Consequently, if the design requirements shift – for instance, the synthesis of a new chemical compound – new round(s) of optimization experiments would be necessary. This requirement for additional experimentation may not align well with demands for rapid customization. In contrast to traditional methods, our integrated workflow that combines collaborative robotics with SVM classification, active learning loops, and data augmentation, resulted in a prediction model that consistently delivers high accuracy across the entire parameter space. Consequently, to respond to

diverse design requests for customization, our prediction model can automatically identify the optimal fabrication parameters, eliminating the need for repeated optimization experiments.

2. Automatic design of conductive aerogels with programmable properties. Through the robotics/ML-integrated workflow, the prediction model can perform two-way design tasks: (1) predicting the physicochemical properties of conductive aerogels based on their fabrication parameters and (2) automating the inverse design of conductive aerogels to align with specific property requirements. The prediction model is well-suited for crafting conductive aerogels with customizable mechanical and electrical properties. Given the laborious nature of self-assembly and the absence of robotic systems available for fully automating freeze drying processes, our prediction model can directly identify viable and optimal fabrication parameters, without the need for iterative optimization experiments. Compared with the state-of-the-art works of conductive aerogels (see **Table S12**), our automatic design method demonstrates superior efficiency in discovering conductive aerogels with programmable properties.

3. Data-driven elucidation of design principles in complex material systems. **Table S13** presents a comparison of our work with multiple studies focusing on MXene-based aerogels, including MXene/CNT, MXene/rGO, MXene/ANF, and MXene/CNF. Traditionally, these state-of-the-art works explore the intricate correlations between fabrication parameters and aerogel properties through the one-factor-at-a-time (OFAT) design of experiment (DoE) method. The OFAT DoE method has yielded diverse design insights across different MXene-based aerogel systems. For example, Zeng *et al.*, Xu *et al.*, Jiang *et al.*, and Wang *et al.* groups discovered that increasing the MXene loading decreased the compressive strength and electrical resistance of conductive aerogels. In contrast, our research harnesses the synergy of collaborative robotics and advanced AI/ML predictive analytics. By constructing a robust prediction model followed by model interpretation (e.g., SHAP), we can investigate several component–property correlations and generalize design principles with statistical significance, offering a more systematic understanding of complex material systems.

Revision Made

Page 28 of the revised manuscript

Compared with the state-of-the-art works of conductive aerogels, our robotics/ML-integrated workflow demonstrates multiple advancements, as summarized in **Table S11**,⁷³⁻⁷⁵ **S12**,^{2,8,19,20,29,33,76-85} and **S13**.^{8,19,20,29,33,76-82} The first advancement is to construct a prediction model with high accuracy across the entire parameter space. As detailed in **Table S11**, the state-of-the-art works typically focus on pinpointing an optimal set of fabrication parameters to maximize device performance or identifying precise conditions necessary for successful chemical reactions. While some of these works also utilize high-throughput robotic platforms to navigate a parameter space with multiple degrees of freedom (DOFs), they do not intend to examine data points that stray from the primary objectives, such as determining the most effective parameter combination or sequence of reaction steps. Consequently, if the design requirements shift – for instance, the synthesis of a new chemical compound – new round(s) of optimization experiments would be necessary. This requirement for additional experimentation may not align well with demands for rapid customization. In contrast to traditional methods, our integrated workflow that combines collaborative robotics with SVM classification, active learning loops, and data augmentation, resulted in a prediction model that consistently delivers high accuracy across the entire parameter

space. Consequently, to respond to diverse design requests for customization, our prediction model can automatically identify the optimal fabrication parameters, eliminating the need for repeated optimization experiments.

The second advancement is to automate the design of conductive aerogels with programmable properties. Through the robotics/ML-integrated workflow, the prediction model can perform two-way design tasks: (1) predicting the physicochemical properties of conductive aerogels based on their fabrication parameters and (2) automating the inverse design of conductive aerogels to align with specific property requirements. The prediction model is well-suited for crafting conductive aerogels with customizable mechanical and electrical properties. Given the laborious nature of self-assembly and the absence of robotic systems available for fully automating freeze drying processes, our prediction model can directly identify viable and optimal fabrication parameters, without the need for iterative optimization experiments. Compared with the state-of-the-art works of conductive aerogels (see **Table S12**), our automatic design method demonstrates superior efficiency in discovering conductive aerogels with programmable properties.

The third advancement is to elucidate data-driven design principles in complex material systems. **Table S13** presents a comparison of our work with multiple studies focusing on MXene-based aerogels, including MXene/CNT, MXene/rGO, MXene/ANF, and MXene/CNF. Traditionally, these state-of-the-art works explore the intricate correlations between fabrication parameters and aerogel properties through the one-factor-at-a-time (OFAT) design of experiment (DoE) method. The OFAT DoE method has yielded diverse design insights across different MXene-based aerogel systems. For example, Zeng *et al.*, Xu *et al.*, Jiang *et al.*, and Wang *et al.* groups discovered that increasing the MXene loading decreased the compressive strength and electrical resistance of conductive aerogels.^{33,76,80,81} In contrast, our research harnesses the synergy of collaborative robotics and advanced AI/ML predictive analytics. By constructing a robust prediction model followed by model interpretation (e.g., SHAP), we can investigate several component–property correlations and generalize design principles with statistical significance, offering a more systematic understanding of complex material systems.

Table S11. Comparison of our AI/ML framework with the state-of-the-art works.

Independent Variables (# of DOFs)	Optimization/Prediction Targets (# of Property Labels)	Multi-Property Optimization	Sampling Method	Experiments Required for New Design Requests	Sample Preparation and Characterization Platforms	Data Collection Rate	Machine Learning Algorithms	Ref.
Fuel-to-oxidizer ratio, fuel blend, total concentration, anneal temperature (4 DOFs)	Determining the optimal fabrication parameters for high-conductivity palladium films at low annealing temperatures (2 property labels)	Yes	Bayesian optimization (w/ qEHVI acquisition function)	Possible	Ada: Precision 4-axis laboratory robot (N9)/6-axis collaborative robot (UR5e)	2 data per hour	Gaussian process regression	1
Column count, column outer radius, column thickness, twisted angle (4 DOFs)	Pinpointing the optimal structural parameters for high compression toughness (1 property label)	No	Bayesian optimization	Possible	BEAR: 3D printers/robotic arm/scale/universal testing machine	64 data per day	Gaussian process regression	2
Reagent injection sequence, reaction time length, volume (>40 DOFs)	Determining the optimal reaction parameters for synthesizing hetero-nanostructures with high λ_{AP} , R_{PV} , I_{PL} (3 property labels)	Yes	Reinforcement learning	Possible	AlphaFlow: A self-driven fluidic lab consisting of reagent injection, droplet oscillation, optical sampling, phase separation, and waste collection.	2 data per hour	Reinforcement learning	3
MXene loading, CNF loading, gelatin loading, GA loading, mixture loading (3 DOFs)	Aiming to predict across the full parameter space to produce conductive aerogels with customized mechanical and electrical properties (2 property labels)	Yes	Active learning (w/ A score acquisition function)	No	OT-2 robot and UR5e-automated compression tester	20 data per 2.5 days	Support-vector machine / artificial neural network / data augmentation	This work

Table S12. Comparison of our robotics/ML-integrated workflow with the state-of-the-art works regarding the production of conductive MXene aerogels.

Design Strategy	Aerogel Composition	Aerogel Density (mg cm^{-3})	σ_{30} (kPa)	Electrical Property ($-$)	Electrical Property Measurement	Ref.
Quaternary Aerogel Systems with MXene/CNF						
Robotics/ML-integrated workflow	MXene/CNF/gelatin/GA	2.0 – 14.0	0.05 – 16.1	$1.4 - 10^{10} \Omega$	Two-electrode testing system	This work
Binary Aerogel Systems with MXene/CNF						
Design of experiment	MXene/CNF	4.0	0.5	$2.8 \times 10^1 \text{ S cm}^{-1}$	Four-point probe	4
		4.0	0.8	$4.0 \times 10^{-1} \text{ S cm}^{-1}$		
Design of experiment	MXene/CNF	50	7.0	$2.6 \times 10^1 \Omega \text{ cm}$	Two-electrode testing system	5
		50	3.4	$1.4 \times 10^1 \Omega \text{ cm}$		
Ternary Aerogel Systems with MXene/CNF						
Design of experiment	MXene/CNF/CNT	7.5	0.4	$2.4 \times 10^3 \text{ S cm}^{-1}$	Four-point probe	6
Design of experiment	MXene/CNF/PMDI	18.0	40.0	$1.1 \times 10^0 \Omega \text{ cm}$	Four-point probe	7
Other MXene Aerogel Systems						
Design of experiment	MTMS treated bacterial cellulose/MXene	6.2	1.6	$1.2 \times 10^2 \text{ S cm}^{-1}$	Four-point probe	8
Design of experiment	MXene/CNT	9.1	0.5	$4.5 \times 10^2 \text{ S cm}^{-1}$	Four-point probe	9

Design Strategy	Aerogel Composition	Aerogel Density (mg cm ⁻³)	σ_{30} (kPa)	Electrical Property (-)	Electrical Property Measurement	Ref.
Design of experiment	PGPDMS/MXene	9.9	0.1	–	–	10
Design of experiment	MXene/rGO	12.2	7.5	3.6×10^1 S cm ⁻¹	Three-electrode setup	11
Design of experiment	MXene	12.6	1.0	5.4×10^1 S cm ⁻¹	Two-electrode testing system	12
Design of experiment	MXene/ANF	25.0	17.0	1.0×10^4 Ω cm	Two-electrode testing system	13
Design of experiment	MXene/CNT/ANF	42.0	2.3	2.7×10^0 Ω cm	Four-point probe	14
		42.0	0.8	2.0×10^{-1} Ω cm	Four-point probe	
Design of experiment	MXene/Waterborne polyurethane (WPU)	–	3.0	–	–	15
Other Conductive Aerogel Systems						
Design of experiment	rGO	2.3	6.0	1.6×10^1 S cm ⁻¹	Four-point probe	16
Design of experiment	rGO	8.5	8.2	4.3×10^1 S cm ⁻¹	Four-electrode testing system	17
Design of experiment	GO/PVA	10.0	5.0	3.0×10^0 S cm ⁻¹	Source meter	18
Design of experiment	GO	25.0	1.0	9.7×10^0 S cm ⁻¹	Source meter	19

Abbreviations

rGO: Reduced graphene oxide

GO: Graphene oxide

MTMS: Methyltrimethoxysilane

CNT: Carbon nanotube

PGPDMS: Glycidoxypropyldimethoxymethylsilane

PVA: Poly(vinyl alcohol)

PMDI: Poly((phenyl isocyanate)-co-formaldehyde)

ANF: Aramid nanofiber

WPU: Waterborne polyurethane

Table S13. Comparison of our AI/ML and data-driven approach with the state-of-the-art works regarding design insight elucidation.

Insight Elucidation Strategy	Building Block(s)	Discovery Method(s)	Design Insight(s)	Ref.
Quaternary Aerogel Systems with MXene/CNF (Our Work)				
AI/ML and data-driven approach	MXene/CNF/gelatin/GA	Spearman's analysis, SHapley Additive exPlanations (SHAP) model interpretation	 1. The mixture loading had the most significant impact on the mechanical properties of conductive aerogels. 2. The MXene loading had the most significant impact on the electrical properties of conductive aerogels. 	This work
Binary Aerogel Systems with MXene/CNF				
Design of experiment	MXene/CNF	Experimental observation	 1. The addition of 33 wt.% CNF led to a 350% increase in the compressive moduli of conductive aerogels. 2. The addition of 17 wt.% CNF had a minimal impact on the conductivity of conductive aerogels, while the addition of 50 wt.% CNF significantly decreased the conductivity of these aerogels. 	4
Design of experiment	MXene/CNF	Experimental observation	 1. Increasing the MXene loading decreased the compressive strength of conductive aerogels under 33% strain. 2. Increasing the MXene loading from 0.7 to 14 wt.% decreased the resistance of conductive aerogels from 26.3 to 13.8 Ω. 	5
Ternary Aerogel Systems with MXene/CNF				
Design of experiment	MXene/CNF/CNT	Experimental observation	 1. The presence of CNF plays a crucial role in maintaining the structural integrity of aerogels. 	6
Design of experiment	MXene/CNF/PMDI	Experimental observation	 1. Adding 30 wt.% CNFs increased the compressive modulus of MXene/CNF aerogels by 155%. 2. The mechanical strength of MXene/CNF aerogels was improved by chemically crosslinking them with PMDI. 	7
Other MXene Aerogel Systems				

Insight Elucidation Strategy	Building Block(s)	Discovery Method(s)	Design Insight(s)	Ref.
Design of experiment	MTMS treated bacterial cellulose/MXene	Experimental observation	 1. Aerogels fabricated from a 1:1 ratio of silylated bacterial cellulose to MXene maintained a residual height of >95% and exhibited a compressive strength of 6 kPa under 50% strain. 2. Conversely, aerogels produced from a 1:1 ratio of untreated bacterial cellulose and MXene retained a residual height of approximately 70% and displayed a compressive strength of 5 kPa. 	8
Design of experiment	MXene/CNT	Experimental observation	 1. An aerogel, produced from a 95:5 ratio of MXene to CNT, exhibited a plastic deformation of 4.2% and an electrical conductivity of 450 S m⁻¹. 2. Another aerogel, produced from a 60:40 ratio of MXene to CNT, showed a plastic deformation of 2.1% and an electrical conductivity of 100 S m⁻¹. 	9
Design of experiment	PGPDMS/MXene	Experimental observation	 1. The integration of PGPDMS into the MXene interlayers produced an ultra-soft aerogel. 2. At aerogel densities of 7.0 and 10 mg cm⁻³, the compressive moduli of the PGPDMS/MXene aerogels were 58 and 140 Pa respectively. 	10
Design of experiment	MXene/rGO	Experimental observation	 1. As the MXene loading increased from 25 to 75 wt.%, the aerogels' maximum strain decreased from 95% to 60%. 2. When the annealing temperature rose from room temperature to 300 °C, both the maximum strain and the conductivity of conductive aerogels increased. 	11
Design of experiment	MXene	Experimental observation	 1. The compressive stress and electrical conductivity of MXene aerogels both increased with the rise in aerogel density. 2. Factors such as sheet alignment, pore size, and overall aerogel microstructure influenced the aerogels' electrochemical properties. 	12
Design of experiment	MXene/ANF	Experimental observation	 1. As the MXene loading increased from 30, 50, to 70 wt.%, the compressive stress of MXene/ANF aerogels decreased by 8.6%, 25.3%, and 30.1%, respectively. 2. The aerogel's resistance also displayed a similar trend in relation to the MXene loading. 	13
Design of experiment	MXene/CNT/ANF	Experimental observation	 1. As the CNT loading increased, the mechanical properties of the composite aerogels decreased. 2. On the other hand, a higher CNT loading improved the electrical conductivity of these composite aerogels. 	14

Insight Elucidation Strategy	Building Block(s)	Discovery Method(s)	Design Insight(s)	Ref.
Design of experiment	MXene/Waterborne polyurethane (WPU)	Experimental observation	 1. The composite aerogel, made from functionalized cellulose nanocrystal (f-NCC), MXene, and polyurethane, withstood 100 compression–relaxation cycles, maintaining 76.2% of the initial maximum stress. 2. The wood-like microstructure, interconnected network, and interactions between the f-NCC, MXene, and PU matrix improved the compressibility and elasticity of the composite aerogels. 	16

Comment 2: Exploration of known correlations: In the introduction, the authors mention that correlations between the ingredients are “largely” unexplored, but do not elaborate on those that are explored. Providing more details on these known correlations and potentially incorporating them into their experimentation could increase efficiency.

Response R2.3

In the revised manuscript, we included several studies that focused on MXene-based aerogels, including MXene/CNT, MXene/rGO, MXene/ANF, and MXene/CNF. As shown in **Table S13**, these state-of-the-art works investigated the complex correlations between fabrication parameters and aerogel properties *via* the OFAT DoE method. Through this conventional approach, different design principles were discovered in various MXene-based aerogel systems. For example, Zeng *et al.*, Xu *et al.*, Jiang *et al.*, and Wang *et al.* groups discovered that increasing the MXene loading decreased the compressive strength and electrical resistance of conductive aerogels. In contrast, our research harnesses the synergy of collaborative robotics and advanced AI/ML predictive analytics. By constructing a robust prediction model followed by model interpretation (e.g., SHAP), we can investigate several component–property correlations and generalize design principles with statistical significance, offering a more systematic understanding of complex material systems.

Comment 3: Comparison with traditional (experimental) approaches: The implementation of SHAP and FEA to elucidate the ML algorithm’s reasoning is well-founded. However, it would be beneficial if the authors provided more context on the advantages of this approach compared to traditional experimental methods, such as Design of Experiments (DoE), optimization methods, and experimentally validated finite element analysis (FEA) models, which might yield comparable results in potentially less time.

Response R2.4

In response to Reviewer #2’s request, we conducted comparative analyses of different sampling methods: random sampling, Latin hypercube sampling, and active learning sampling (as demonstrated in our study). **Fig. S13** and **Table S6** provide details on the three sampling cycles for each method, with five separate physical experiments carried out in each cycle. As shown in **Fig. S14a**, the active learning sampling method outperformed the others, achieving a success rate of >95% in recommending the MXene/CNF/gelatin/GA ratios that resulted in the production of A-grade aerogels. In comparison, random sampling and Latin hypercube sampling yielded lower success rates of 80% and 67%, respectively.

After the three sampling cycles, we applied the UIP method to the real data points collected from the different sampling methods and synthesized virtual data points at a ratio of 1-to-1,000. Using these real and virtual data points, we trained multiple ANN-based prediction models. As shown in **Fig. S14b**, among all the sampling methods, the prediction model trained on data points from the active learning sampling exhibited superior learning efficacy and enhanced prediction accuracy, achieving the lowest recorded MAE value of 1.1 kPa. Whereas the MAEs from random sampling and Latin hypercube sampling were 5.8 kPa and 5.0 kPa, respectively, after completing three cycles.

Revision Made

In addition, the selection of a suitable sampling method is important for the construction of an accurate prediction model. Comparative analyses of different sampling methods were conducted: random sampling, Latin hypercube sampling, and active learning sampling (as demonstrated in our study). **Fig. S13** and **Table S6** provide details on the three sampling cycles for each method, with five separate physical experiments carried out in each cycle. As shown in **Fig. S14a**, the active learning sampling method outperformed the others, achieving a success rate of >95% in recommending the MXene/CNF/gelatin/GA ratios that resulted in the production of *A*-grade aerogels. In comparison, random sampling and Latin hypercube sampling yielded lower success rates of 80% and 67%, respectively. After the three sampling cycles, we applied the UIP method to the real data points collected from the different sampling methods and synthesized virtual data points at a ratio of 1-to-1,000. Using these real and virtual data points, we trained multiple ANN-based prediction models. As shown in **Fig. S14b**, among all the sampling methods, the prediction model trained on data points from the active learning sampling exhibited superior learning efficacy and enhanced prediction accuracy, achieving the lowest recorded MAE values of 1.1 kPa. Whereas the MAEs from random sampling and Latin hypercube sampling were 5.8 kPa and 5.0 kPa, respectively, after completing three cycles.

Page 16 of Supporting Information

Fig. S13. Distribution profiles of data points collected from different sampling methods. (a) Random sampling, (b) Latin hypercube sampling, and (c) active learning sampling. A total of 3 sampling cycles were performed, and 5 physical experiments were conducted in each cycle.

Table S6. Training dataset for various prediction models based on different sampling methods.

Sampling Method	MXene (wt. %)	CNF (wt. %)	Gelatin (wt. %)	Mixture Loading (mg mL ⁻¹)	Grade
Random	67.2	28.5	4.3	7.5	B
	1.0	46.7	52.4	10.0	B
	32.3	38.8	28.9	10.0	B
	53.4	46.6	0.0	2.5	A
	52.9	2.1	45.0	7.5	D
	11.1	54.6	34.3	2.5	C
	39.9	56.3	3.8	2.5	A
	8.6	46.5	44.9	7.5	A
	30.8	19.2	50.0	2.5	C
	42.3	29.6	28.1	7.5	A
	56.8	32.1	11.1	10.0	A
	46.3	33.7	19.9	2.5	A
	21.0	38.5	40.4	7.5	A
	28.6	45.7	25.7	5.0	A
	44.9	25.5	29.6	10.0	A
Latin Hypercube	13.5	43.9	42.7	7.5	B
	65.1	25.3	9.6	7.5	A
	1.3	51.8	46.9	5.0	C
	37.1	27.3	35.6	5.0	B
	13.2	41.6	45.2	2.5	D
	39.9	21.1	38.9	2.5	C
	53.2	20.8	26.0	2.5	B
	40.8	10.6	48.6	2.5	C
	33.5	39.8	26.7	10.0	B
	13.3	77.2	9.6	7.5	B
	89.1	5.3	5.6	10.0	B
	42.6	28.7	28.7	7.5	B
	23.5	21.9	54.6	7.5	B
	24.7	43.2	32.1	10.0	B
31.1	30.7	38.1	7.5	C	
Active Learning	12.4	40.1	47.4	10.0	A
	33.3	35.6	31.1	10.0	A
	11.2	76.5	12.3	10.0	A
	71.0	11.5	17.4	10.0	A
	33.3	54.1	12.7	10.0	A
	13.1	27.0	59.9	7.5	A

Sampling Method	MXene (wt. %)	CNF (wt. %)	Gelatin (wt. %)	Mixture Loading (mg mL⁻¹)	Grade
	33.3	35.6	31.1	7.5	A
	11.2	76.5	12.3	7.5	A
	20.2	67.9	11.9	7.5	A
	12.4	40.1	47.4	5.0	A
	13.1	27.0	59.9	5.0	A
	60.6	27.7	11.7	5.0	A
	33.3	54.1	12.7	5.0	A
	11.2	76.5	12.3	2.5	A
	33.3	54.1	12.7	2.5	A

Fig. S14. Prediction model performance based on different sampling methods. (a) Quantity of conductive aerogels at different grades based on different sampling methods. The active learning sampling was able to recommend the MXene/CNF/gelatin ratios with a >95% successful rate in producing *A*-grade aerogels. (b) MAE values of the prediction models based on different sampling methods. Trained by the data points collected from the active learning sampling, the prediction model demonstrated better learning efficiency and higher prediction accuracy, as evidenced by the lowest MAE of 1.1 kPa. Whereas the MAEs from random sampling and Latin hypercube sampling were 5.8 kPa and 5.0 kPa, respectively, after completing three cycles.

Comment 4: Quantitative comparison with DoE: The authors briefly mention DoE in the conclusion, claiming greater efficiency for their approach. This assertion would be more compelling if supported by quantitative data, such as the number of experimental runs required by a DoE design to achieve a similar level of information and predictive capability.

Response R2.5

To compare our ML framework with the one-factor-at-a-time (OFAT) design of experiment (DoE) method, we calculated the number of data points needed to cover the entire design space using **Equations S1** and **S2**,

$$S_x = \frac{(n!)}{k \cdot (x-1)!} \quad (\text{S1})$$

$$n = x - 1 + k \quad (\text{S2})$$

, where S_x denotes the number of data points, x is the number of DOFs, and k is the step size. The Python code is available on GitHub at <https://github.com/oshin71/Programmable-MXene-Aerogel/blob/main>. Using the step size of 2.0 wt.%, the OFAT DoE method requires 5,300 data points. As depicted in **Fig. S6**, the number of data points is inversely correlated with the step size.

Revision Made

Page 6 of the revised manuscript

To establish a comprehensive database linking the fabrication parameters with the end properties of conductive aerogels, over 5,300 data points are required, given a step size of 2.0 wt.% and four mixture loadings (see our estimation in **Note S2** and **Fig. S6**).

Page 29 of Supporting Information

Note S2. Estimated number of experiments required to build an extensive dataset for conductive aerogels.

To compare our ML framework with the one-factor-at-a-time (OFAT) design of experiment (DoE) method, we calculated the number of data points needed to cover the entire design space using **Equations S1** and **S2**,

$$S_x = \frac{(n!)}{k! \cdot (x-1)!} \quad (\text{S1})$$

$$n = x - 1 + k \quad (\text{S2})$$

, where S_x denotes the number of data points, x is the number of DOFs, and $1/k$ is the step size. The Python code is available on GitHub at <https://github.com/oshin71/Programmable-MXene-Aerogel/blob/main>.

For the composition labels, three DOFs ($x = 3$) are identified for the fabrication of conductive aerogels. For the loading label of conductive aerogels, one DOF is identified. Considering a step size of 2.0 wt.% for the MXene, CNF, and gelatin loadings ($\frac{1}{k} = 0.02$), two options (+/-) for the GA incorporation, and four mixture loadings, a total number of experiments required to construct an extensive dataset is estimated to be 5,300. In addition, two property labels, including σ_{30} and R_0 , need to be collected for each conductive aerogel. Using the step size of 2.0

wt.%, the OFAT DoE method requires 5,300 data points. As depicted in **Fig. S6**, the number of data points is inversely correlated with the step size.

Comment 5: Writing improvements: Address minor grammatical issues, such as inconsistencies in parenthesis usage (e.g., “silk nanofibrils, chitosan”). Also, avoid repetitive language, like the repetition of “conductivity” in the very first sentence. Some statements are vague, such as “Multiple blind tests were performed by different researchers to maintain unbiased evaluations.” Specifying the number of tests would provide clarity.

Response R2.6

We appreciate the detailed review from Reviewer #2. We have carefully revised both the manuscript and Supporting Information to correct the grammatical errors. Also, we have specified the number of blind tests (i.e., two blind tests) to be “two” and revised the sentence into “Two blind tests were performed by different researchers to maintain unbiased evaluations”.

Revision Made

Page 3 of the revised manuscript

Conductive aerogels have gained significant research interests due to their ultralight characteristics, adjustable mechanical properties, and outstanding electrical performance.¹⁻⁶

Page 3 of the revised manuscript

Key building blocks include conductive nanomaterials like carbon nanotubes, graphene, $Ti_3C_2T_x$ MXene nanosheets,²⁴⁻³⁰ functional fillers like cellulose nanofibers (CNFs), silk nanofibrils, and chitosan,^{29,31-34} polymeric binders like gelatin,^{25,26} and crosslinking agents that include glutaraldehyde (GA) and metal ions.^{30,35-37}

Page 7 of the revised manuscript

Two blind tests were performed by different researchers to maintain unbiased evaluations.

Comment 6: Methodological details: Clarify how the freeze-drying process is automated, if at all. While there is some context in the Supporting Information, a brief mention in the main document would be beneficial.

Response R2.7

We apologize for any confusion caused by our previous descriptions. In the revised manuscript, we have incorporated **Fig. S1** in the revised manuscript, which provides a clear depiction of the integrated human–machine teaming workflow. It is important to note that there are no robotic platforms that fully automate the entire freeze drying process. We have automated the preparation of aqueous mixtures and the execution of compression tests through the use of an OT-2 robot and a UR5e-automated compression tester, respectively.

Revision Made

Page 4 of the revised manuscript

Herein, we developed an integrated platform that combines the capabilities of collaborative robots with AI/ML predictions to accelerate the design of conductive aerogels with programmable mechanical and electrical properties (see **Fig. S1** for the robot–human teaming workflow).

Page 4 of Supporting Information

Fig. S1. Robot–human teaming workflow for producing conductive aerogels. There are no robotic platforms that fully automate the entire freeze drying process. We have automated the preparation of aqueous mixtures and the execution of compression tests through the use of an OT-2 robot and a UR5e-automated compression tester, respectively.

Comment 7: Clarification on testing procedures: The results suggest that conductivity was tested after compressive stress testing. It would be helpful if the authors clarified whether new samples were produced for each test or if the same samples were reused (Page 11).

Response R2.8

We apologize that our previous descriptions of the testing procedures may not have been adequately clear. To clarify, in our experiments, separate conductive aerogel samples were designated for each type of test. A sample subjected to electrical resistance testing was not employed again for compression testing, and *vice versa*. It ensured that each test was conducted on an uncompromised conductive aerogel, preserving the integrity of our results.

Revision Made

Page 38 of the revised manuscript

Electrical resistance measurements of conductive aerogels. The initial electrical resistance, R_0 , of the conductive aerogels were measured using a two-electrode system connected with an electrochemical workstation (Autolab PGSTAT302N, Metrohm) or an Industrial Multimeter

(Fluke 87V). The gap between two electrodes was fixed at 1 cm. A sample subjected to electrical resistance testing was not employed again for compression testing, and *vice versa*. It ensured that each test was conducted on an uncompromised conductive aerogel, preserving the integrity of our results.

Reviewer #3

This manuscript reported the conductive hydrogels inks with different components by machine learning and freeze drying. The resultant aerogels with the desired conductivity and compressibility are obtained by designing different ratios.

Response R3.1

We would like to express our gratitude to Reviewer #3 for dedicating his/her time to evaluate our work and provide valuable comments. We have carefully considered all of the comments and revised the manuscript and Supporting Information accordingly.

However, it is common to prepare both conductive and elastic aerogels by introducing elastic CNF and conductive MXene, many studies have explored MXene/CNF systems including their conductivity and compressibility. In this experiment, machine learning only replaced the feeding and matching process, but could not independently analyze the results to improve the formula.

Comment 1: MXene/CNF aerogels have been widely reported, and optimization schemes can be selected based on the establishment of conductive thresholds and elastic systems.

Response R3.2

We appreciate that Reviewer #3 for providing his/her comments. We would like to provide three perspectives to highlight the novelty of our work.

In the revised manuscript, we have compared our robotics/ML-integrated workflow with the state-of-the-art works, as summarized in **Table S11, S12, and S13**.

1. Construction of a prediction model with high accuracy across the entire parameter space. Our work successfully demonstrated a versatile workflow that combined collaborative robotics and AI/ML predictions, enabling efficient development and optimization of various functional nanocomposites (e.g., conductive aerogels) with user-designated properties. By incorporating collaborative robotic technologies, we can significantly enhance the speed of acquiring high-quality data points and minimize human variations. Meanwhile, the collection of data points for the next rounds of experiments can be guided by the AI/ML predictions.

As detailed in **Table S11**, the state-of-the-art works typically focus on pinpointing an optimal set of fabrication parameters to maximize device performance or identifying precise conditions necessary for successful chemical reactions. While some of these works also utilize high-throughput robotic platforms to navigate a parameter space with multiple degrees of freedom (DOFs), they do not intend to examine data points that stray from the primary objectives, such as determining the most effective parameter combination or sequence of reaction steps. Consequently, if the design requirements shift – for instance, the synthesis of a new chemical compound – new round(s) of optimization experiments would be necessary. This requirement for additional experimentation may not align well with demands for rapid customization. In contrast to traditional methods, our integrated workflow that combines collaborative robotics with SVM classification, active learning loops, and data augmentation, resulted in a prediction model that consistently delivers high accuracy across the entire parameter space. Consequently, to respond to diverse design requests for customization, our prediction model can automatically identify the optimal fabrication parameters, eliminating the need for repeated optimization experiments.

2. Automatic design of conductive aerogels with programmable properties. Through the robotics/ML-integrated workflow, the prediction model can perform two-way design tasks: (1) predicting the physicochemical properties of conductive aerogels based on their fabrication parameters and (2) automating the inverse design of conductive aerogels to align with specific property requirements. The prediction model is well-suited for crafting conductive aerogels with customizable mechanical and electrical properties. Given the laborious nature of self-assembly and the absence of robotic systems available for fully automating freeze drying processes, our prediction model can directly identify viable and optimal fabrication parameters, without the need for iterative optimization experiments. Compared with the state-of-the-art works of conductive aerogels (see **Table S12**), our automatic design method demonstrates superior efficiency in discovering conductive aerogels with programmable properties.

3. Data-driven elucidation of design principles in complex material systems. **Table S13** presents a comparison of our work with multiple studies focusing on MXene-based aerogels, including MXene/CNT, MXene/rGO, MXene/ANF, and MXene/CNF. Traditionally, these state-of-the-art works explore the intricate correlations between fabrication parameters and aerogel properties through the one-factor-at-a-time (OFAT) design of experiment (DoE) method. The OFAT DoE method has yielded diverse design insights across different MXene-based aerogel systems. For example, Zeng *et al.*, Xu *et al.*, Jiang *et al.*, and Wang *et al.* groups discovered that increasing the MXene loading decreased the compressive strength and electrical resistance of conductive aerogels. In contrast, our research harnesses the synergy of collaborative robotics and advanced AI/ML predictive analytics. By constructing a robust prediction model followed by model interpretation (e.g., SHAP), we can investigate several component–property correlations and generalize design principles with statistical significance, offering a more systematic understanding of complex material systems.

Revision Made

Page 28 of the revised manuscript

Compared with the state-of-the-art works of conductive aerogels, our robotics/ML-integrated workflow demonstrates multiple advancements, as summarized in **Table S11**,⁷³⁻⁷⁵ **S12**,^{2,8,19,20,29,33,76-85} and **S13**.^{8,19,20,29,33,76-82} The first advancement is to construct a prediction model with high accuracy across the entire parameter space. As detailed in **Table S11**, the state-of-the-art works typically focus on pinpointing an optimal set of fabrication parameters to maximize device performance or identifying precise conditions necessary for successful chemical reactions. While some of these works also utilize high-throughput robotic platforms to navigate a parameter space with multiple degrees of freedom (DOFs), they do not intend to examine data points that stray from the primary objectives, such as determining the most effective parameter combination or sequence of reaction steps. Consequently, if the design requirements shift – for instance, the synthesis of a new chemical compound – new round(s) of optimization experiments would be necessary. This requirement for additional experimentation may not align well with demands for rapid customization. In contrast to traditional methods, our integrated workflow that combines collaborative robotics with SVM classification, active learning loops, and data augmentation, resulted in a prediction model that consistently delivers high accuracy across the entire parameter space. Consequently, to respond to diverse design requests for customization, our prediction model can automatically identify the optimal fabrication parameters, eliminating the need for repeated optimization experiments.

The second advancement is to automate the design of conductive aerogels with programmable properties. Through the robotics/ML-integrated workflow, the prediction model can perform two-way design tasks: (1) predicting the physicochemical properties of conductive aerogels based on their fabrication parameters and (2) automating the inverse design of conductive aerogels to align with specific property requirements. The prediction model is well-suited for crafting conductive aerogels with customizable mechanical and electrical properties. Given the laborious nature of self-assembly and the absence of robotic systems available for fully automating freeze drying processes, our prediction model can directly identify viable and optimal fabrication parameters, without the need for iterative optimization experiments. Compared with the state-of-the-art works of conductive aerogels (see **Table S12**), our automatic design method demonstrates superior efficiency in discovering conductive aerogels with programmable properties.

The third advancement is to elucidate data-driven design principles in complex material systems. **Table S13** presents a comparison of our work with multiple studies focusing on MXene-based aerogels, including MXene/CNT, MXene/rGO, MXene/ANF, and MXene/CNF. Traditionally, these state-of-the-art works explore the intricate correlations between fabrication parameters and aerogel properties through the one-factor-at-a-time (OFAT) design of experiment (DoE) method. The OFAT DoE method has yielded diverse design insights across different MXene-based aerogel systems. For example, Zeng *et al.*, Xu *et al.*, Jiang *et al.*, and Wang *et al.* groups discovered that increasing the MXene loading decreased the compressive strength and electrical resistance of conductive aerogels.^{33,76,80,81} In contrast, our research harnesses the synergy of collaborative robotics and advanced AI/ML predictive analytics. By constructing a robust prediction model followed by model interpretation (e.g., SHAP), we can investigate several component–property correlations and generalize design principles with statistical significance, offering a more systematic understanding of complex material systems.

Table S11. Comparison of our AI/ML framework with the state-of-the-art works.

Independent Variables (# of DOFs)	Optimization/Prediction Targets (# of Property Labels)	Multi-Property Optimization	Sampling Method	Experiments Required for New Design Requests	Sample Preparation and Characterization Platforms	Data Collection Rate	Machine Learning Algorithms	Ref.
Fuel-to-oxidizer ratio, fuel blend, total concentration, anneal temperature (4 DOFs)	Determining the optimal fabrication parameters for high-conductivity palladium films at low annealing temperatures (2 property labels)	Yes	Bayesian optimization (w/ qEHVI acquisition function)	Possible	Ada: Precision 4-axis laboratory robot (N9)/6-axis collaborative robot (UR5e)	2 data per hour	Gaussian process regression	1
Column count, column outer radius, column thickness, twisted angle (4 DOFs)	Pinpointing the optimal structural parameters for high compression toughness (1 property label)	No	Bayesian optimization	Possible	BEAR: 3D printers/robotic arm/scale/universal testing machine	64 data per day	Gaussian process regression	2
Reagent injection sequence, reaction time length, volume (>40 DOFs)	Determining the optimal reaction parameters for synthesizing hetero-nanostructures with high λ_{AP} , R_{PV} , I_{PL} (3 property labels)	Yes	Reinforcement learning	Possible	AlphaFlow: A self-driven fluidic lab consisting of reagent injection, droplet oscillation, optical sampling, phase separation, and waste collection.	2 data per hour	Reinforcement learning	3
MXene loading, CNF loading, gelatin loading, GA loading, mixture loading (3 DOFs)	Aiming to predict across the full parameter space to produce conductive aerogels with customized mechanical and electrical properties (2 property labels)	Yes	Active learning (w/ A score acquisition function)	No	OT-2 robot and UR5e-automated compression tester	20 data per 2.5 days	Support-vector machine / artificial neural network / data augmentation	This work

Table S12. Comparison of our robotics/ML-integrated workflow with the state-of-the-art works regarding the production of conductive MXene aerogels.

Design Strategy	Aerogel Composition	Aerogel Density (mg cm^{-3})	σ_{30} (kPa)	Electrical Property ($-$)	Electrical Property Measurement	Ref.
Quaternary Aerogel Systems with MXene/CNF						
Robotics/ML-integrated workflow	MXene/CNF/gelatin/GA	2.0 – 14.0	0.05 – 16.1	$1.4 - 10^{10} \Omega$	Two-electrode testing system	This work
Binary Aerogel Systems with MXene/CNF						
Design of experiment	MXene/CNF	4.0	0.5	$2.8 \times 10^1 \text{ S cm}^{-1}$	Four-point probe	4
		4.0	0.8	$4.0 \times 10^{-1} \text{ S cm}^{-1}$		
Design of experiment	MXene/CNF	50	7.0	$2.6 \times 10^1 \Omega \text{ cm}$	Two-electrode testing system	5
		50	3.4	$1.4 \times 10^1 \Omega \text{ cm}$		
Ternary Aerogel Systems with MXene/CNF						
Design of experiment	MXene/CNF/CNT	7.5	0.4	$2.4 \times 10^3 \text{ S cm}^{-1}$	Four-point probe	6
Design of experiment	MXene/CNF/PMDI	18.0	40.0	$1.1 \times 10^0 \Omega \text{ cm}$	Four-point probe	7
Other MXene Aerogel Systems						
Design of experiment	MTMS treated bacterial cellulose/MXene	6.2	1.6	$1.2 \times 10^2 \text{ S cm}^{-1}$	Four-point probe	8
Design of experiment	MXene/CNT	9.1	0.5	$4.5 \times 10^2 \text{ S cm}^{-1}$	Four-point probe	9

Design Strategy	Aerogel Composition	Aerogel Density (mg cm ⁻³)	σ_{30} (kPa)	Electrical Property (-)	Electrical Property Measurement	Ref.
Design of experiment	PGPDMS/MXene	9.9	0.1	–	–	10
Design of experiment	MXene/rGO	12.2	7.5	3.6×10^1 S cm ⁻¹	Three-electrode setup	11
Design of experiment	MXene	12.6	1.0	5.4×10^1 S cm ⁻¹	Two-electrode testing system	12
Design of experiment	MXene/ANF	25.0	17.0	1.0×10^4 Ω cm	Two-electrode testing system	13
Design of experiment	MXene/CNT/ANF	42.0	2.3	2.7×10^0 Ω cm	Four-point probe	14
		42.0	0.8	2.0×10^{-1} Ω cm	Four-point probe	
Design of experiment	MXene/Waterborne polyurethane (WPU)	–	3.0	–	–	15
Other Conductive Aerogel Systems						
Design of experiment	rGO	2.3	6.0	1.6×10^1 S cm ⁻¹	Four-point probe	16
Design of experiment	rGO	8.5	8.2	4.3×10^1 S cm ⁻¹	Four-electrode testing system	17
Design of experiment	GO/PVA	10.0	5.0	3.0×10^0 S cm ⁻¹	Source meter	18
Design of experiment	GO	25.0	1.0	9.7×10^0 S cm ⁻¹	Source meter	19

Abbreviations

rGO: Reduced graphene oxide

GO: Graphene oxide

MTMS: Methyltrimethoxysilane

CNT: Carbon nanotube

PGPDMS: Glycidoxypropyldimethoxymethylsilane

PVA: Poly(vinyl alcohol)

PMDI: Poly((phenyl isocyanate)-co-formaldehyde)

ANF: Aramid nanofiber

WPU: Waterborne polyurethane

Table S13. Comparison of our AI/ML and data-driven approach with the state-of-the-art works regarding design insight elucidation.

Insight Elucidation Strategy	Building Block(s)	Discovery Method(s)	Design Insight(s)	Ref.
Quaternary Aerogel Systems with MXene/CNF (Our Work)				
AI/ML and data-driven approach	MXene/CNF/gelatin/GA	Spearman's analysis, SHapley Additive exPlanations (SHAP) model interpretation	 1. The mixture loading had the most significant impact on the mechanical properties of conductive aerogels. 2. The MXene loading had the most significant impact on the electrical properties of conductive aerogels. 	This work
Binary Aerogel Systems with MXene/CNF				
Design of experiment	MXene/CNF	Experimental observation	 1. The addition of 33 wt.% CNF led to a 350% increase in the compressive moduli of conductive aerogels. 2. The addition of 17 wt.% CNF had a minimal impact on the conductivity of conductive aerogels, while the addition of 50 wt.% CNF significantly decreased the conductivity of these aerogels. 	4
Design of experiment	MXene/CNF	Experimental observation	 1. Increasing the MXene loading decreased the compressive strength of conductive aerogels under 33% strain. 2. Increasing the MXene loading from 0.7 to 14 wt.% decreased the resistance of conductive aerogels from 26.3 to 13.8 Ω. 	5
Ternary Aerogel Systems with MXene/CNF				
Design of experiment	MXene/CNF/CNT	Experimental observation	 1. The presence of CNF plays a crucial role in maintaining the structural integrity of aerogels. 	6
Design of experiment	MXene/CNF/PMDI	Experimental observation	 1. Adding 30 wt.% CNFs increased the compressive modulus of MXene/CNF aerogels by 155%. 2. The mechanical strength of MXene/CNF aerogels was improved by chemically crosslinking them with PMDI. 	7
Other MXene Aerogel Systems				

Insight Elucidation Strategy	Building Block(s)	Discovery Method(s)	Design Insight(s)	Ref.
Design of experiment	MTMS treated bacterial cellulose/MXene	Experimental observation	 1. Aerogels fabricated from a 1:1 ratio of silylated bacterial cellulose to MXene maintained a residual height of >95% and exhibited a compressive strength of 6 kPa under 50% strain. 2. Conversely, aerogels produced from a 1:1 ratio of untreated bacterial cellulose and MXene retained a residual height of approximately 70% and displayed a compressive strength of 5 kPa. 	8
Design of experiment	MXene/CNT	Experimental observation	 1. An aerogel, produced from a 95:5 ratio of MXene to CNT, exhibited a plastic deformation of 4.2% and an electrical conductivity of 450 S m⁻¹. 2. Another aerogel, produced from a 60:40 ratio of MXene to CNT, showed a plastic deformation of 2.1% and an electrical conductivity of 100 S m⁻¹. 	9
Design of experiment	PGPDMS/MXene	Experimental observation	 1. The integration of PGPDMS into the MXene interlayers produced an ultra-soft aerogel. 2. At aerogel densities of 7.0 and 10 mg cm⁻³, the compressive moduli of the PGPDMS/MXene aerogels were 58 and 140 Pa respectively. 	10
Design of experiment	MXene/rGO	Experimental observation	 1. As the MXene loading increased from 25 to 75 wt.%, the aerogels' maximum strain decreased from 95% to 60%. 2. When the annealing temperature rose from room temperature to 300 °C, both the maximum strain and the conductivity of conductive aerogels increased. 	11
Design of experiment	MXene	Experimental observation	 1. The compressive stress and electrical conductivity of MXene aerogels both increased with the rise in aerogel density. 2. Factors such as sheet alignment, pore size, and overall aerogel microstructure influenced the aerogels' electrochemical properties. 	12
Design of experiment	MXene/ANF	Experimental observation	 1. As the MXene loading increased from 30, 50, to 70 wt.%, the compressive stress of MXene/ANF aerogels decreased by 8.6%, 25.3%, and 30.1%, respectively. 2. The aerogel's resistance also displayed a similar trend in relation to the MXene loading. 	13
Design of experiment	MXene/CNT/ANF	Experimental observation	 1. As the CNT loading increased, the mechanical properties of the composite aerogels decreased. 2. On the other hand, a higher CNT loading improved the electrical conductivity of these composite aerogels. 	14

Insight Elucidation Strategy	Building Block(s)	Discovery Method(s)	Design Insight(s)	Ref.
Design of experiment	MXene/Waterborne polyurethane (WPU)	Experimental observation	 1. The composite aerogel, made from functionalized cellulose nanocrystal (f-NCC), MXene, and polyurethane, withstood 100 compression–relaxation cycles, maintaining 76.2% of the initial maximum stress. 2. The wood-like microstructure, interconnected network, and interactions between the f-NCC, MXene, and PU matrix improved the compressibility and elasticity of the composite aerogels. 	16

Comment 2: Machine learning replaces the human formulation of different proportions of the mixture, the lack of a reasonable screening scheme - analysis - self-correction - the process of drawing conclusions.

Response R3.3

In response to Reviewer 3's comment, we have further elaborated on our integrated workflow, which combines robotics with machine learning to enhance the data-driven process of material development. Our innovative workflow consists of four main phases: screening, analysis, design/optimization, and self-correction/validation.

In the screening phase, we used an automated pipetting robot (i.e., OT-2) to efficiently prepare 264 different mixtures with varying ratios of MXene/CNF/gelatin/GA and mixture loadings, replacing laborious manual experimentation. After the freeze drying process, we assessed the structural integrity of conductive aerogels to train a SVM classifier, which played a critical role in defining a feasible parameter space with high DOFs. **In the analysis phase**, we avoided the traditional one-factor-at-a-time (OFAT) method and instead employed a more robust strategy involving 8 active learning cycles and data augmentation. During active learning, we fabricated and characterized 162 different types of conductive aerogels, uncovering intricate correlations between fabrication parameters and aerogel properties. Afterward, we used a data augmentation technique to enrich our dataset, enabling the construction of an ANN-based prediction model with high accuracy. **In the design/optimization phase**, we leveraged the prediction model to perform two-way design tasks: (1) predicting the physicochemical properties of conductive aerogels based on fabrication parameters, and (2) automating the inverse design of conductive aerogels to meet specific property criteria. **In the self-correction/validation phase**, our prediction model surpassed traditional experience-based design approaches by conducting clustering analyses to identify optimal fabrication parameters that aligned with targeted design requests. Furthermore, we used model interpretation and finite element simulations to validate the established correlations.

Meanwhile, we would like to emphasize that the integration of collaborative robotics and AI/ML predictions into the research workflow aims to enhance, rather than replace, the expertise of researchers. While our robotics/ML-integrated system aims to reduce the experimental burden and improve efficiency, the domain knowledge and experience of materials, researchers remain crucial in collecting high-quality data points and constructing a prediction model.

In our work, the expertise of researchers was indispensable at several pivotal points. First, after using the OT-2 robot to prepare a library of conductive aerogels, it was imperative for researchers to grade each aerogel based on its structural integrity. This critical evaluation largely relied on the judgment of experienced researchers and ensured a robust screening process. Subsequently, when establishing the SVM model, the determination of the *A*-grade probability threshold was another point where researchers' insights were important. Setting this threshold was a balancing act. If the threshold was set too high, the design space would become constricted, potentially inhibiting the innovation of high-performance conductive aerogels. If the threshold was set too low, it would likely result in an unfavorable rate of experimental failure, impeding the efficiency of data collection in active learning cycles. Moreover, the data curation phase was essential for the development of an accurate predictive model. As evidenced in **Fig. S7**, incorporating complete stress-strain curves of conductive aerogels into the training dataset led to a notable decrease in prediction accuracy, as reflected by the MAE value increased to 4.5 kPa. The

selection of suitable property labels for model training was therefore contingent upon the researchers' deep understanding and domain expertise. Finally, the data augmentation phase was informed by researchers' empirical observations, as depicted in **Fig. S11**. These observations were foundational for the generation of virtual data points, which were enhanced with Gaussian noise to simulate real-world variability. Through each of these phases, the contribution of researchers was crucial, not only in applying their experiential knowledge for qualitative assessments but also in exercising their domain expertise to guide the quantitative modeling and data analysis.

In addition, the selection of a suitable sampling method is important for the construction of an accurate prediction model. Comparative analyses of different sampling methods were conducted: random sampling, Latin hypercube sampling, and active learning sampling (as demonstrated in our study). **Fig. S13** and **Table S6** provide details on the three sampling cycles for each method, with five separate physical experiments carried out in each cycle. As shown in **Fig. S14a**, the active learning sampling method outperformed the others, achieving a success rate of >95% in recommending the MXene/CNF/gelatin/GA ratios that resulted in the production of *A*-grade aerogels. In comparison, random sampling and Latin hypercube sampling yielded lower success rates of 80% and 67%, respectively. After the three sampling cycles, we applied the UIP method to the real data points collected from the different sampling methods and synthesized virtual data points at a ratio of 1-to-1,000. Using these real and virtual data points, we trained multiple ANN-based prediction models. As shown in **Fig. S14b**, among all the sampling methods, the prediction model trained on data points from the active learning sampling exhibited superior learning efficacy and enhanced prediction accuracy, achieving the lowest recorded MAE value of 1.1 kPa. after completing the three cycles. Whereas the MAEs from random sampling and Latin hypercube sampling were 5.8 kPa and 5.0 kPa, respectively, after completing three cycles.

Revision Made

Page 16 of the revised manuscript

In addition, the selection of a suitable sampling method is important for the construction of an accurate prediction model. Comparative analyses of different sampling methods were conducted: random sampling, Latin hypercube sampling, and active learning sampling (as demonstrated in our study). **Fig. S13** and **Table S6** provide details on the three sampling cycles for each method, with five separate physical experiments carried out in each cycle. As shown in **Fig. S14a**, the active learning sampling method outperformed the others, achieving a success rate of >95% in recommending the MXene/CNF/gelatin/GA ratios that resulted in the production of *A*-grade aerogels. In comparison, random sampling and Latin hypercube sampling yielded lower success rates of 80% and 67%, respectively. After the three sampling cycles, we applied the UIP method to the real data points collected from the different sampling methods and synthesized virtual data points at a ratio of 1-to-1,000. Using these real and virtual data points, we trained multiple ANN-based prediction models. As shown in **Fig. S14b**, among all the sampling methods, the prediction model trained on data points from the active learning sampling exhibited superior learning efficacy and enhanced prediction accuracy, achieving the lowest recorded MAE values of 1.1 kPa. Whereas the MAEs from random sampling and Latin hypercube sampling were 5.8 kPa and 5.0 kPa, respectively, after completing three cycles.

Page 33 of the revised manuscript

In conclusion, an integrated workflow, which combined automated robotic experiments with an AI/ML framework, consisted of four main phases: screening, analysis, design/optimization, and self-correction/validation. In the screening phase, we used an automated pipetting robot (i.e., OT-2) to efficiently prepare 264 different mixtures with varying ratios of MXene/CNF/gelatin/GA and mixture loadings, replacing laborious manual experimentation. After the freeze drying process, we assessed the structural integrity of conductive aerogels to train a SVM classifier, which played a critical role in defining a feasible parameter space with high DOFs. In the analysis phase, we avoided the traditional OFAT method and instead employed a more robust strategy involving 8 active learning cycles and data augmentation. During active learning, we fabricated and characterized 162 different types of conductive aerogels, uncovering intricate correlations between fabrication parameters and aerogel properties. Afterward, we used a data augmentation technique to enrich our dataset, enabling the construction of an ANN-based prediction model with high accuracy. In the design/optimization phase, we leveraged the prediction model to perform two-way design tasks: (1) predicting the physicochemical properties of conductive aerogels based on fabrication parameters, and (2) automating the inverse design of conductive aerogels to meet specific property criteria. In the self-correction/validation phase, our prediction model surpassed traditional experience-based design approaches by conducting clustering analyses to identify optimal fabrication parameters that aligned with targeted design requests. Furthermore, we used model interpretation and FE simulations to validate the established correlations. Finally, the model's predictive power was harnessed to discover a strain-insensitive conductive aerogel with compatible mechanical strength, high electrical conductivity, and ultralow pressure sensitivity suitable for wearable heating applications. The successful integration of robotic experiments, AI/ML predictions, and FE simulations has demonstrated a synergistic approach, with potential

applications beyond conductive aerogels, such as tactile sensors,^{86,87} stretchable conductors,^{88,89} catalysts,^{59,60} and electrochemical electrolyte optimization.^{57,58}

Meanwhile, we would like to emphasize that the integration of collaborative robotics and AI/ML predictions into the research workflow aims to enhance, rather than replace, the expertise of researchers. While our robotics/ML-integrated system aims to reduce the experimental burden and improve efficiency, the domain knowledge and experience of materials researchers remain crucial in collecting high-quality data points and constructing a prediction model. In our work, the expertise of researchers was indispensable at several pivotal points. First, after using the OT-2 robot to prepare a library of conductive aerogels, it was imperative for researchers to grade each aerogel based on its structural integrity. This critical evaluation largely relied on the judgment of experienced researchers and ensured a robust screening process. Subsequently, when establishing the SVM model, the determination of the *A*-grade probability threshold was another point where researchers' insights were important. Setting this threshold was a balancing act. If the threshold was set too high, the design space would become constricted, potentially inhibiting the innovation of high-performance conductive aerogels. If the threshold was set too low, it would likely result in an unfavorable rate of experimental failure, impeding the efficiency of data collection in active learning cycles. Moreover, the data curation phase was essential for the development of an accurate predictive model. As evidenced in **Fig. S7**, incorporating complete stress–strain curves of conductive aerogels into the training dataset led to a notable decrease in prediction accuracy, as reflected by the MAE value increased to 4.5 kPa. The selection of suitable property labels for model training was therefore contingent upon the researchers' deep understanding and domain expertise. Finally, the data augmentation phase was informed by researchers' empirical observations, as depicted in **Fig. S11**. These observations were foundational for the generation of virtual data points, which were enhanced with Gaussian noise to simulate real-world variability. Through each of these phases, the contribution of researchers was crucial, not only in applying their experiential knowledge for qualitative assessments but also in exercising their domain expertise to guide the quantitative modeling and data analysis.

Fig. S18. Positive relationships between aerogel density and mixture loading. We characterized the densities of conductive aerogels at four different MXene/CNF/gelatin/GA ratios and four mixture loadings, which ranged from 2.5 to 10.0 mg mL⁻¹. Data are presented as mean \pm s.d., $n = 3$, with each independent experiment marked by an open black circle.

Fig. S19. Dependence of electrical resistances and aerogel microstructures on mixture loading. (a) Correlations between aerogel electrical resistance and mixture loading. (b) SEM images of conductive aerogels at the MXene/CNF/gelatin/GA ratio of 11/77/12/+ at different mixture loadings. (c) SEM images of conductive aerogels at the MXene/CNF/gelatin/GA ratio of 61/28/11/- at different mixture loadings. (d) SEM images of conductive aerogels at the MXene/CNF/gelatin/GA ratio of 43/42/15/- at different mixture loadings.

Fig. S13. Distribution profiles of data points collected from different sampling methods. (a) Random sampling, (b) Latin hypercube sampling, and (c) active learning sampling. A total of 3 sampling cycles were performed, and 5 physical experiments were conducted in each cycle.

Table S6. Training dataset for various prediction models based on different sampling methods.

Sampling Method	MXene (wt. %)	CNF (wt. %)	Gelatin (wt. %)	Mixture Loading (mg mL ⁻¹)	Grade
Random	67.2	28.5	4.3	7.5	B
	1.0	46.7	52.4	10.0	B
	32.3	38.8	28.9	10.0	B
	53.4	46.6	0.0	2.5	A
	52.9	2.1	45.0	7.5	D
	11.1	54.6	34.3	2.5	C
	39.9	56.3	3.8	2.5	A
	8.6	46.5	44.9	7.5	A
	30.8	19.2	50.0	2.5	C
	42.3	29.6	28.1	7.5	A
	56.8	32.1	11.1	10.0	A
	46.3	33.7	19.9	2.5	A
	21.0	38.5	40.4	7.5	A
	28.6	45.7	25.7	5.0	A
	44.9	25.5	29.6	10.0	A
Latin Hypercube	13.5	43.9	42.7	7.5	B
	65.1	25.3	9.6	7.5	A
	1.3	51.8	46.9	5.0	C
	37.1	27.3	35.6	5.0	B
	13.2	41.6	45.2	2.5	D
	39.9	21.1	38.9	2.5	C
	53.2	20.8	26.0	2.5	B
	40.8	10.6	48.6	2.5	C
	33.5	39.8	26.7	10.0	B
	13.3	77.2	9.6	7.5	B
	89.1	5.3	5.6	10.0	B
	42.6	28.7	28.7	7.5	B
	23.5	21.9	54.6	7.5	B
	24.7	43.2	32.1	10.0	B
31.1	30.7	38.1	7.5	C	
Active Learning	12.4	40.1	47.4	10.0	A
	33.3	35.6	31.1	10.0	A
	11.2	76.5	12.3	10.0	A
	71.0	11.5	17.4	10.0	A
	33.3	54.1	12.7	10.0	A
	13.1	27.0	59.9	7.5	A

Sampling Method	MXene (wt. %)	CNF (wt. %)	Gelatin (wt. %)	Mixture Loading (mg mL⁻¹)	Grade
	33.3	35.6	31.1	7.5	A
	11.2	76.5	12.3	7.5	A
	20.2	67.9	11.9	7.5	A
	12.4	40.1	47.4	5.0	A
	13.1	27.0	59.9	5.0	A
	60.6	27.7	11.7	5.0	A
	33.3	54.1	12.7	5.0	A
	11.2	76.5	12.3	2.5	A
	33.3	54.1	12.7	2.5	A

Fig. S14. Prediction model performance based on different sampling methods. (a) Quantity of conductive aerogels at different grades based on different sampling methods. The active learning sampling was able to recommend the MXene/CNF/gelatin ratios with a >95% successful rate in producing A-grade aerogels. (b) MAE values of the prediction models based on different sampling methods. Trained by the data points collected from the active learning sampling, the prediction model demonstrated better learning efficiency and higher prediction accuracy, as evidenced by the lowest MAE of 1.1 kPa. Whereas the MAEs from random sampling and Latin hypercube sampling were 5.8 kPa and 5.0 kPa, respectively, after completing three cycles.

Comment 3: Why choose MXene/CNF system, how to solve the oxidation problem and the aggregation problem introduced by CNF of MXene?

Response R3.4

Thank you for your insightful comments. We would like to draw your attention to **Note S1** in the original manuscript, which provides further details on the selection criteria for MXene nanosheets, CNFs, and gelatin for the fabrication of our conductive aerogels. These building blocks were chosen due to their synergistic properties to enhance the overall performance of the aerogels through synergistic effects. Together, these building blocks have great potential to create a nanocomposite with optimized electrical conductivity, mechanical strength, and structural integrity.

As shown in **Fig. S4a,b**, the zeta potentials of MXene and CNF dispersions were monitored before and after two-week of storage. Their zeta potentials remained consistent, with no signs of oxidation or aggregation detected.

Moreover, the Ti 2p XPS spectra provided in **Fig. 10** reveal that the characteristic Ti–C bonds indicative of MXene integrity are preserved, as there is no evidence of new Ti–C bond formation. This suggests that the MXene nanosheets maintain their structural integrity throughout the aerogel fabrication process, even with the GA introduction. The XPS finding confirms that the intrinsic properties of the MXene nanosheets remained intact after the aerogel fabrication processes.

Revision Made

Page 4 of the revised manuscript

To produce various conductive aerogels, four building blocks were selected, including MXene nanosheets, CNFs, gelatin, and GA crosslinker (see **Note S1** and **Fig. S2** for the selection rationale and model expansion strategy).

Page 6 of the revised manuscript

As shown in **Fig. S4a,b**, the zeta potentials of MXene and CNF dispersions were monitored before and after two-week of storage. Their zeta potentials remained consistent, with no signs of oxidation or aggregation detected.

Page 8 of the revised manuscript

Moreover, the Ti 2p XPS spectra provided in **Fig. S10** reveal that the characteristic Ti–C bonds indicative of MXene integrity are preserved, as there is no evidence of new Ti–C bond formation. This suggests that the MXene nanosheets maintain their structural integrity throughout the aerogel fabrication process, even with the GA introduction. The XPS finding confirms that the intrinsic properties of the MXene nanosheets remained intact after the aerogel fabrication processes.

Fig. S4. Fabrication of conductive aerogels with tunable compositions and mixture loadings. (a) Zeta potentials of MXene before and after one-week storage. Data are presented as mean \pm s.d., $n = 3$, with each independent experiment marked by an open black circle. (b) Zeta potentials of CNF dispersions before and after one-week storage. Data are presented as mean \pm s.d., $n = 3$, with each independent experiment marked by an open black circle. (c) Photo of various MXene/CNF/gelatin/GA mixtures with various ratios and mixture loadings as well as the resulting conductive aerogels.

Fig. S10. Ti 2p spectra of MXene/CNF aerogels without and with GA incorporation. (a) Ti 2p spectrum of the conductive aerogel without GA incorporation (at the MXene/CNF/gelatin/GA ratio of 80/20/0/- and the mixture loading of 10 mg mL⁻¹). **(b)** Ti 2p spectrum of the conductive aerogel with GA incorporation (at the MXene/CNF/gelatin/GA ratio of 80/20/0/+ and the mixture loading of 10 mg mL⁻¹).

Note S1. Rationale of building block selection and model expansion strategy.

The rationale behind selecting $\text{Ti}_3\text{C}_2\text{T}_x$ MXene nanosheets, CNFs, and gelatin for the fabrication of conductive aerogels results from the synergistic effects of these building blocks on the aerogel's properties.

1. $\text{Ti}_3\text{C}_2\text{T}_x$ MXene nanosheets:

- Electrical conductivity: $\text{Ti}_3\text{C}_2\text{T}_x$ MXene nanosheets, belonging to a class of two-dimensional materials, are known for their high electrical conductivity, which makes them a preferred choice for boosting the electrical conductivity of conductive aerogels.
- Surface chemistry: With abundant functional groups like $-\text{OH}$, $-\text{F}$, and $-\text{O}$ on the MXene surfaces, MXene nanosheets can form stable interactions with other building blocks (such as CNFs and gelatin) and enhance the structural stability of the resulting aerogels.

2. CNFs:

- Structural reinforcement: CNFs are renowned for their superior mechanical strength and stiffness, acting as reinforcement units in the conductive aerogels.
- High aspect ratio: The high aspect ratio of CNFs can facilitate the formation of a percolated network throughout the conductive aerogel, leading to enhanced mechanical resilience.
- Hydrophilicity: The inherent hydrophilicity of CNFs aids in the uniform distribution of other building blocks within the aerogel structure.

3. Gelatin:

- Gelling agent: Gelatin, a natural biopolymer, excels as a gelling agent, especially when combined with crosslinking agents, such as glutaraldehyde.
- Affinity to other building blocks: Because of its ability to form robust molecular interactions, gelatin can bridge MXenes and CNFs effectively, creating potential hybrid systems of conductive aerogels.

The selection of building blocks is pivotal for defining the functional capabilities of lightweight aerogels. For instance, it is advantageous to integrate more natural components into the parameter space of multi-component aerogels, such as gellan gum, polyvinyl alcohol (PVA), and sodium alginate. When composed at varying ratios, these aerogels can be rehydrated to create a variety of functional hydrogels suitable for agricultural applications, including serving as growth media for microgreens and baby greens (see **Fig. S2a,b**). To enhance the prediction model with a broader range of building blocks, it is advisable to adopt a model expansion method to augment its predictive power. As shown in **Fig. S2c**, additional active learning loops should be performed under the guidance of the prediction model. During the model expansion phase, additional experiments are required to refine the SVM classifier and to update the ANN-based model. By strategically selecting new components in tandem with the model expansion method, the prediction model can consistently enlarge its parameter space and broadened the range of achievable functions. However, this model expansion incurs additional active learning cycles, leading to higher time and cost implications. To mitigate this, the development of automated fabrication and characterization platforms is essential to enhance data collection efficiency and lessen human labor demands. Additionally, establishing an open database would offer substantial benefits to the wider research community.

Fig. S2. Schematic illustration of a model expansion strategy. Photos of microgreens grown on hydrogel media (a) without and (b) without the nanofillers. (c) Workflow of a model expansion method.

REVIEWERS' COMMENTS

Reviewer #1 (Remarks to the Author):

This work incorporated the robotics-accelerated experimentation, machine learning, and simulation to accelerate the tailored design and application of conductive aerogels. This work is novel, and the authors provide sufficient details and results to support their conclusions and claims. I recommend publishing this manuscript in Nature Communications.

Reviewer #2 (Remarks to the Author):

I am satisfied with and appreciate the changes the authors have made to the manuscript, which I believe have significantly improved its quality. I recommend the publication of the article without further revisions.

Reviewer #3 (Remarks to the Author):

After careful review, I find the manuscript has been significantly improved according my questions, and it can be published as is.